# A Computationally Efficient Sparsified Online Newton Method

**Devvrit** [*][†]
Department of Computer Science
The University of Texas at Austin
devvrit.03@gmail.com

**Sai Surya Duvvuri**[*]
Department of Computer Science
The University of Texas at Austin
subramanyamdvss@gmail.com

**Rohan Anil**
Google DeepMind
rohananil@google.com

**Vineet Gupta**
Google
vineet@google.com

**Cho-Jui Hsieh**
CS Department, UCLA & Google
chohsieh@cs.ucla.edu

**Inderjit Dhillon**
Google
isd@google.com

## Abstract

Second-order methods hold significant promise for enhancing the convergence of deep neural network training; however, their large memory and computational demands have limited their practicality. Thus there is a need for scalable second-order methods that can efficiently train large models. In this paper, we introduce the Sparsified Online Newton (SONew) method, a memory-efficient second-order algorithm that yields a sparsified yet effective preconditioner. The algorithm emerges from a novel use of the LogDet matrix divergence measure; we combine it with sparsity constraints to minimize regret in the online convex optimization framework. Empirically, we test our method on large scale benchmarks of up to 1B parameters. We achieve up to 30% faster convergence, 3.4% relative improvement in validation performance, and 80% relative improvement in training loss, in comparison to memory efficient optimizers including first order methods. Powering the method is a surprising fact – imposing structured sparsity patterns, like tridiagonal and banded structure, requires little to no overhead, making it as efficient and parallelizable as first-order methods. In wall-clock time, tridiagonal SONew is only about 3% slower per step than first-order methods but gives overall gains due to much faster convergence. In contrast, one of the state-of-the-art (SOTA) memory-intensive second-order methods, Shampoo, is unable to scale to large benchmarks. Additionally, while Shampoo necessitates significant engineering efforts to scale to large benchmarks, SONew offers a more straightforward implementation, increasing its practical appeal. SONew code is available at: https://github.com/devvrit/SONew

## 1 Introduction

Stochastic first order methods which use the negative gradient direction to update parameters have become the standard for training deep neural networks (DNNs). Gradient-based preconditioning involves finding an update direction, by multiplying the gradient with a preconditioner matrix carefully chosen from gradients observed in previous iterations, to improve convergence. (Full-

---

[*]equal contribution, [†] Work done while at Google

37th Conference on Neural Information Processing Systems (NeurIPS 2023).

matrix) Adagrad [15], online Newton method [25] and natural gradient descent [3] use a full-matrix preconditioner, but computing and storing the full matrix is infeasible when there are millions of parameters. Thus, diagonal versions such as diagonal Adagrad, Adam [33], and RMSprop [28] are now widely used to train DNNs due to their scalability.

Several higher-order methods have previously been applied to deep learning ([24, 5, 23, 38]). All these methods use Kronecker product factorizations that reduce computational and storage costs to make them feasible for training neural networks. However, to precondition a $d_1 \times d_2$ parameter matrix, these methods require matrix inverse operations, which take $\mathcal{O}(d_1^3 + d_2^3)$ time and $\mathcal{O}(d_1^2 + d_2^2)$ space. In comparison, first-order methods use $\mathcal{O}(d_1 d_2)$ time and memory, which is linear in the number of parameters. For instance, when $d_1 = k d_2$, the memory used by Shampoo, $d_1^2 + d_2^2$ floating point numbers is $\mathcal{O}(k)$ times the number of parameters, which could be arbitrarily large depending on $k$. This calls for further research in developing efficient second-order optimization techniques to train DNNs with memory and time complexity linear in the number of parameters.

In this paper, we present a novel Sparsified Online Newton (SONew) method, which only requires linear time and space complexity, to train large-scale DNNs. We derive the algorithm through two steps, classical regret analysis followed by a sparsification step. In more detail, regret analysis when using a preconditioner reveals that the error is bounded by two terms, the first depends on the change in the preconditioning matrix, while the second depends on the generalized gradient norm (see Section 3 for more details). We take a novel approach of minimizing the second term while regularizing two successive preconditioners to be close in the LogDet matrix divergence measure [34] (see Section 3 for the intuition behind choosing LogDet divergence). This analysis naturally yields us an Online Newton method [25]. To make it computationally efficient, we further sparsify the preconditioner by finding a sparse approximation that is close in LogDet divergence. Thus we are consistent in using the same measure (LogDet divergence) in both the regularization and sparsification steps. This gives us our SONew method, which achieves linear complexity by leveraging structured sparsity patterns, such as tridiagonal and banded, in the preconditioner. This is unlike most existing online Newton methods that require quadratic space and cubic time complexity. By making each step linear time, the SONew method can be applied to train modern DNNs as efficiently as first order methods. Further, our method is embarrassingly parallelizable thus making negligible the overhead of computing the preconditioner. We also show that introducing sparsity allows us to reduce the condition number of the problem dynamically to improve numerical stability.

We strengthen the relationship between sparse LogDet divergence minimization and online convex optimization by establishing an optimal $\mathcal{O}(\sqrt{T})$ regret upper bound for tridiagonal sparsity pattern. In our experiments on an MLP Autoencoder and Graph Neural Network (GNN), we found that our method outperformed first-order methods in terms of training loss within the same training time, while Shampoo (second-order method) takes significantly longer. In our experiments on Vision Transformers on Imagenet and GNN on OGBG-molpcba, we achieve a target validation performance using 10% and 30% fewer iterations respectively compared to Adam, the SOTA optimizer for both benchmarks. Furthermore, using the same number of iterations as Adam we observe 0.7% and 3.4% relative improvement for ViT and GNN respectively in validation performance. From an optimization point of view, SONew achieves 9% and 80% better relative training loss for ViT and GNN respectively. It is worth noting that Shampoo statistics required $\sim 7 \times \#params$ for ViT whereas tridiag-SONew uses only $2 \times \#params$ for its statistics. We also test another recently proposed memory efficient second order optimizer, rfdSON [37], but found its performance suboptimal to the best performing first order method. Owing to SONew's scalability, we train a Large Language Model (LLM) with 1 billion parameters and compare it with AdaFactor [45], a popularly used first order optimizer to train LLMs [11]. SONew achieves the same performance as AdaFactor using $26\%$ fewer steps, resulting in a $1.35\times$ faster training. When using the same number of steps, SONew obtained a $1.7\%$ relative better train loss. In terms of implementation, SONew is just a few lines of code (Equation (13)) without complex engineering challenges, rendering it even more useful and practical.

## 2   Background

The inner product between matrices is defined as $\langle A, B \rangle = \mathrm{Tr}(A^T B)$, where $\mathrm{Tr}(.)$ denotes the matrix trace. The Frobenius norm of a matrix $A$ is $\|A\|_F = \sqrt{\mathrm{Tr}(A^T A)}$, while its spectral norm is $\|A\|_2 = \max_x \|Ax\|_2 / \|x\|_2$. We use $I_n \in \mathbb{R}^{n \times n}$ to denote an identity matrix. We use $S_n, S_n^{++}$ to denote the set of symmetric, and positive definite matrices respectively. The generalized norm of a

vector $x \in \mathbb{R}^n$ with respect to matrix $A \in S_n^{++}$ is defined as $\|x\|_A = \sqrt{x^T A x}$. We use $\det(A)$ to denote the determinant of matrix $A$, and $\operatorname{diag}(A)$ to denote the diagonal matrix with $\operatorname{diag}(A)_{ii} = A_{ii}$. We use $\mathcal{G}$ and $\tilde{\mathcal{G}}$ to denote a graph and its sub-graph with a vertex set $[n] = \{1, \ldots, n\}$. Let $E_{\mathcal{G}}$ denote set of edges in graph $\mathcal{G}$, and $\operatorname{neig}_{\mathcal{G}}(i)$ denote neighbours of vertex $i$ in graph $\mathcal{G}$. A sparse symmetric matrix $A \in \mathbb{R}^{n \times n}$ follows a sparsity structure graph $\mathcal{G}$ if $A_{i,j} = 0 \; \forall (i,j) \notin E_{\mathcal{G}}$, . Note that set of all such matrices form a linear subspace. We use $S_n(\mathcal{G})^{++}$ to denote the set of positive definite matrices with sparsity structure given by graph $\mathcal{G}$, i.e, if $X \in S_n(\mathcal{G})^{++}$, then $X_{ij} = 0$ $\forall (i,j) \notin E(\mathcal{G})$. $S_n(\mathcal{G})^{++}$ is an open convex set. Given an index set $I = \{i_1, i_2, .., i_n\}$, we use $A_{II}$ to denote the corresponding principal sub-matrix of $A$.

## 2.1 LogDet matrix divergence

Let $\phi : S_n^{++} \to \mathbb{R}$ be a strictly convex, differentiable function. The Bregman matrix divergence between $X, Y \in S_n^{++}$ is defined as [8, 34]: $\mathrm{D}_\phi(X, Y) = \phi(X) - \phi(Y) - \operatorname{Tr}(\nabla \phi(Y)^T(X - Y))$. Since $\phi$ is convex, $\mathrm{D}_\phi(X, Y) \geq 0$ for all $X, Y \in S_n^{++}$. For example if $\phi(X) = \|X\|_F^2$, the corresponding Bregman divergence $\mathrm{D}_\phi(X, Y) = \|X - Y\|_F^2$ is the squared Frobenius distance. In this paper, we extensively use the convex function $\phi(X) = -\log \det(X)$; the corresponding divergence measure $\mathrm{D}_{\ell\mathrm{d}}(X, Y)$ is called the *LogDet matrix divergence*:

$$\mathrm{D}_{\ell\mathrm{d}}(X, Y) = -\log \det\left(XY^{-1}\right) + \operatorname{Tr}(XY^{-1}) - n. \tag{1}$$

The LogDet divergence is scale invariant to invertible matrices $A$, i.e. $\mathrm{D}_{\ell\mathrm{d}}(A^T X A, A^T Y A) = \mathrm{D}_{\ell\mathrm{d}}(X, Y)$. LogDet divergence can be written in terms of eigendecompositions of $X = V\Sigma V^T$ and $Y = U\Theta U^T$ [34]:

$$\mathrm{D}_{\ell\mathrm{d}}(X, Y) = \sum_i \sum_j (v_i^T u_j)^2 (\sigma_i/\theta_j - \log(\sigma_i/\theta_j) - 1). \tag{2}$$

These two properties are later used in Section 3 to highlight the significance of LogDet divergence in our algorithm.

# 3 SONew: Sparsified Online Newton Method

We now present our proposed algorithm SONew.

## 3.1 Regret minimization via LogDet divergence

We set up our problem under the online convex optimization framework (OCO) [43, 26], where at each round the learner makes a prediction $w_t$ and receives a convex loss $f_t(w_t)$ and gradient $g_t = \nabla f_t(w_t)$ as feedback. The goal of the learner is to reduce regret $R_T$ by predicting $w_t$ so that a low aggregate loss $\sum_{t=1}^T f_t(w_t)$ is achieved compared to the best possible, $w^* = \arg\min_w \sum_{t=1}^T f_t(w)$. Formally, regret is given by

$$R_T(w_1, \ldots, w_T) = \sum_{t=1}^T f_t(w_t) - \sum_{t=1}^T f_t(w^*).$$

Using [10], $R$ regret in online setting yields $R/T$ convergence rate in the stochastic setting. To upper bound this regret, we proceed as in [26] by analyzing the error in the iterates for the update $w_{t+1} := w_t - \eta X_t g_t$, where $X_t \in \mathbb{R}^{n \times n}$. Then $\|w_{t+1} - w^*\|_{X_t^{-1}}^2 = \|w_t - \eta X_t g_t - w^*\|_{X_t^{-1}}^2 = \|w_t - w^*\|_{X_t^{-1}}^2 + \eta^2 g_t^T X_t g_t - 2\eta(w_t - w^*)^T g_t$. The convexity of $f_t$ implies that $f_t(w_t) - f_t(w^*) \leq (w_t - w^*)^T g_t$ leading to $f_t(w_t) - f_t(w^*) \leq \frac{1}{2\eta}(\|w_t - w^*\|_{X_t^{-1}}^2 - \|w_{t+1} - w^*\|_{X_t^{-1}}^2 + \eta^2 g_t^T X_t g_t)$. Summing over all $t \in [T]$ and rearranging reveals the following upper bound on overall regret:

$$R_T \leq \frac{1}{2\eta} \|w_1 - w^*\|_{X_1^{-1}}^2 + \frac{\eta}{2} \sum_{t=1}^T g_t^T X_t g_t + \frac{1}{2\eta} \sum_{t=2}^T (w_t - w^*)^T (X_t^{-1} - X_{t-1}^{-1})(w_t - w^*). \tag{3}$$

Since $w^*$ is unknown, finding $X_t$ which minimizes (3) is infeasible. So to minimize regret, we attempt to minimize the second term in (3) while regularizing $X_t^{-1}$ to be "close" to $X_{t-1}^{-1}$. The nearness measure we choose is the LogDet matrix divergence, thus leading to the following objective

$$X_t = \arg\min_{X \in S_n^{++}} g_t^T X g_t, \text{ such that } \mathrm{D}_{\ell\mathrm{d}}\left(X, X_{t-1}\right) \leq c_t, \tag{4}$$

where $\mathrm{D}_{\ell\mathrm{d}}$ is as in (1). Why do we use the LogDet divergence? From (2), due to the term $\lambda_i/\theta_j$, $\mathrm{D}_{\ell\mathrm{d}}(X, X_{t-1})$ prioritizes matching the smaller eigenvalues of $X_{t-1}$ with those of $X$, i.e., matching the larger eigenvalues of $X_{t-1}^{-1}$ and $X^{-1}$. As a consequence, LogDet divergence regularizes $X$ by matching up its large eigenvalues with those of $X_{t-1}$. For example if smallest and largest eigenvalue of $X_{t-1}$ are $\theta_n$ and $\theta_1$, then for an eigenvalue $\sigma$ of $X$, when $\sigma > \theta_n, \theta_1$, the penalty from (2) for $\theta_n$ is higher than for $\theta_1$, $(\sigma/\theta_n - \log(\sigma/\theta_n) - 1) > (\sigma/\theta_1 - \log(\sigma/\theta_1) - 1)$. This intuition leads us to formulate (4) as our objective. We recall that there is precedence of using the LogDet divergence in the optimization literature; indeed the celebrated BFGS algorithm [9, 17, 22, 44] can be shown to be the unique solution obtained when the LogDet divergence between successive preconditioners, subject to a secant constraint, is minimized (as shown in the 4-page paper by [18]).

The optimization problem in (4) is convex in $X$ since the LogDet divergence is convex in its first argument. The Lagrangian $\mathcal{L}(X, \lambda_t) = g_t^T X g_t + \lambda_t(\mathrm{D}_{\ell\mathrm{d}}(X, X_{t-1}) - c_t) = \mathrm{Tr}(X g_t g_t^T) + \lambda_t(-\log\det\left(XX_{t-1}^{-1}\right) + \mathrm{Tr}(XX_{t-1}^{-1}) - n)) - \lambda_t c_t$. Setting $\nabla\mathcal{L}(X, \lambda_t) = 0$, and using the fact that $\nabla\log\det(X) = X^{-1}$ we get the following update rule:

$$X_t^{-1} = X_{t-1}^{-1} + g_t g_t^T/\lambda_t. \tag{5}$$

We *emphasize* that the update rule (5) arises naturally from our novel use of LogDet divergence to minimize the regret. Moreover, Equation (5) can be seen as a general update rule applicable to numerous existing optimizers. For example, setting $c_t = 0$ (equivalently $\lambda_t = \infty$) $\forall t \in [n]$ in (4) results in no change to the preconditioner in any round. In this case, with $X_0 = I_n$, we get online gradient descent [54]. On the other hand, setting $\lambda_t = 1$ gives the update rule of the online Newton method [25]. Our update rule differs from (full-matrix) Adagrad [15] which has $X_t^{-2} = X_{t-1}^{-2} + g_t g_t^T$.

Maintaining and updating $X_t$ as in (5) is possible by using Sherman-Morrison formula but requires $\mathcal{O}(n^2)$ storage and time complexity. This becomes impractical when $n$ is in the order of millions which is typically the case in DNNs.

### 3.2 Sparsifying the Preconditioner

To minimize the memory needed for maintaining and updating $X_t$ using (5), we adopt the strategy of sparsifying the preconditioner. For existing optimizers such as (full-matrix) Adagrad or the Online Newton method, it is unclear how to sparsify a given preconditioner. Specifically, there is no intuitive approach to assessing the quality of a sparse preconditioner compared to a full-matrix preconditioner. However, since our update rule (5) originates from using LogDet divergence in the regret bound analysis, it gives us a natural metric to measure the quality of a sparse preconditioner. Let's consider the following problem: find a sparse positive definite $X$ with $\|X\|_0 \leq \alpha n$, $\alpha > 1$, such that the objective $\mathrm{D}_{\ell\mathrm{d}}(X, (X_{t-1}^{-1} + g_t g_t^T/\lambda_t)^{-1})$ is minimized. Essentially, this problem imposes a sparsity constraint while requiring the sparse preconditioner to remain close to the full-matrix preconditioner in terms of LogDet divergence.

Due to the $L_0$-norm constraint, this is a non-convex problem, which makes it difficult to solve exactly. Since $L_1$-norm serves as a convex relaxation for the $L_0$ norm, we could use it instead, resulting in the following optimization problem also known as graphical lasso estimator [19]:

$$\min_{X \in S_n^{++}} \mathrm{D}_{\ell\mathrm{d}}\left(X, (X_{t-1}^{-1} + g_t g_t^T/\lambda_t)^{-1}\right) + \gamma \|X\|_1.$$

However, the time taken to solve the above problem, even with the current best methods [7, 29, 16, 53], can still be too large (as these methods take several minutes for a matrix of size million), making it impractical to embed in DNN training.

In this paper, we take a different direction where we use fixed sparsity pattern constraints, specified by a fixed undirected graph $\mathcal{G}$. To sparsify the solution in (5), we formulate the subproblem

$$X_t = \arg\min_{X \in S_n(\mathcal{G})^{++}} \mathrm{D}_{\ell\mathrm{d}}\left(X, (X_{t-1}^{-1} + g_t g_t^T/\lambda_t)^{-1}\right), \tag{6}$$

where $S_n(\mathcal{G})^{++}$ denotes the set of positive definite matrices with the fixed sparsity pattern corresponding to the adjacency matrix of graph $\mathcal{G}$. Note that both steps (4) and (6) use the same LogDet measure.

Owing to the structure of LogDet divergence, (6) can be surprisingly solved in $\mathcal{O}(n)$ and easily parallelizable, for certain sparsity structures $\mathcal{G}$. Algorithm 1 and 2 presents an instantiation of the proposed SONew method, which solves (6) using $\mathcal{O}(n)$ time and memory for banded matrices with band size $b$. In particular a tridiagonal matrix, corresponding to a chain graph, is a banded matrix with bandsize 1.

---

**Algorithm 1** Sparsified Online Newton (SONew) Algorithm

---

    **Inputs**: $\lambda_t :=$ coefficient in the update (10),
    $\mathcal{G} :=$ sparsity graph (banded/tridiagonal),
    $\epsilon :=$ damping parameter,
    $T :=$ total number of iterations/mini-batches,
    $\eta_t :=$ step size/learning rate.
    **Output**: $w_{T+1}$
1:  $H_0 = \epsilon I_d$, $w_1 = 0$
2:  **for** $t \in \{1, \ldots, T\}$ **do**
3:     compute $g_t = \nabla f_t(w_t)$
4:     $H_t := H_{t-1} + P_{\mathcal{G}}(g_t g_t^T / \lambda_t) \in S_n(\mathcal{G})$ with $P_{\mathcal{G}}$ as in (8).    ▷ $\mathcal{O}(n)$ time & memory
5:     Get $L, D =$ SPARSIFIED_INVERSE $(H_t, \mathcal{G})$, where $X_t = LDL^T$ solves (11).
6:     Compute descent direction $u_t = LDL^T g_t$,
7:     $w_{t+1} = w_t - \eta_t u_t$
8:  **end for**
9:  **return** $w_{T+1}$

---

**Algorithm 2** SPARSIFIED_INVERSE$(H, \mathcal{G})$ in $\mathcal{O}(n)$ flops

---

    **Inputs**:$H \in S_n(\mathcal{G})$, is as (10).
    $\mathcal{G} :=$ the banded graph of band size $b \ll n$
    **Outputs**: lower triangular banded $L \in \mathbb{R}^{n \times n}$
    and diagonal matrix $D \in \mathbb{R}^{n \times n}$
1:  **function** SPARSIFIED_INVERSE$(H, \mathcal{G})$
2:     $L := 0$, $D := 0$
3:     $L_{jj} := 1, \forall j \in [n]$
4:     **for** $j \in \{1, \ldots, n\}$ **do**    ▷ parallelizable
5:         Let $H_{jI_j}$ and $H_{I_jI_j}$ be defined as in Section 2, where $I_j = \{j+1, \ldots, j+b\} \cap [n]$,
6:         Solve for $L_{I_jj}$ in the linear system $H_{I_jI_j}L_{I_jj} = -H_{I_jj}$    ▷ $\mathcal{O}(b^3)$ time.
7:         $D_{jj} := 1/(H_{jj} + H_{I_jj}^T L_{I_jj})$
8:     **end for**
9:     **return** $L, D$
10:  **end function**

---

**Maintaining $H_t \in S_n(\mathcal{G})$ in line 4**. Solving the subproblem in (6) naively is impractical since $X_{t-1}^{-1}$ is a dense matrix. However, the structure of the LogDet divergence comes to the rescue; the optimization problem in (6) can be expanded as follows:

$$\underset{X \in S_n(\mathcal{G})^{++}}{\arg\min} -\log \det(X) + \mathrm{Tr}(X(X_{t-1}^{-1} + g_t g_t^T / \lambda_t)). \tag{7}$$

Let us define the projection onto $S_n(\mathcal{G})$, $P_{\mathcal{G}} : \mathbb{R}^{n \times n} \to \mathbb{R}^{n \times n}$ as:

$$P_{\mathcal{G}}(M)_{ij} = \begin{cases} M_{ij} & \text{if } (i,j) \in E_{\mathcal{G}}, \\ 0 & \text{otherwise.} \end{cases} \tag{8}$$

Note that the $\mathrm{Tr}(.)$ term in (7) is dependent only on the non-zero elements of $X \in S_n(\mathcal{G})^{++}$, since $\mathrm{Tr}(AB) = \langle A, B \rangle$, for symmetric matrices $A$ and $B$. Hence, (7) can be written as

$$\underset{X \in S_n(\mathcal{G})^{++}}{\arg\min} -\log \det(X) + \langle X, P_{\mathcal{G}}(X_{t-1}^{-1} + g_t g_t^T / \lambda_t) \rangle, \tag{9}$$

Computing the entire matrix $X_{t-1}^{-1}$ can be avoided by analyzing the optimality condition of (9). Let $g(X) = -\log \det(X) + \langle X, P_{\mathcal{G}}(X_{t-1}^{-1} + g_t g_t^T / \lambda_t) \rangle$ denote the objective function in (9), then the optimality condition of (9) is $P_{\mathcal{G}}(\nabla g(X)) = P_{\mathcal{G}}(\nabla(-\log \det(X) + \langle X, P_{\mathcal{G}}(X_{t-1}^{-1} + g_t g_t^T / \lambda_t)) \rangle) = 0$, since gradients with respective nonzero entries of $X$ should be zero, $\frac{\partial g(X)}{\partial X_{i,j}} = (\nabla_X(g(X)))_{i,j} = 0$, $\forall (i,j) \in E_{\mathcal{G}}$. Using $\nabla(-\log \det(X)) = -X^{-1}$, $\nabla_X(\langle X, Y \rangle) = Y$, and setting $X = X_t$ gives:

$$P_{\mathcal{G}}(X_t^{-1}) - P_{\mathcal{G}}(X_{t-1}^{-1} + g_t g_t^T / \lambda_t) = 0,$$
$$H_t = H_{t-1} + P_{\mathcal{G}}(g_t g_t^T / \lambda_t), \quad \text{where } H_t = P_{\mathcal{G}}(X_t^{-1}) \tag{10}$$

Thus we only need to maintain $H_t = P_{\mathcal{G}}(X_t^{-1})$. This matrix is updated as $H_t = H_{t-1} + P_{\mathcal{G}}(g_t g_t^T / \lambda_t)$. Since $H_t \in S_n(\mathcal{G})$, the update can be done in $\mathcal{O}(|E_{\mathcal{G}}|)$ memory and time, while computing the matrix $X_t^{-1}$ would have cost $\mathcal{O}(n^2)$. In SONew (Algorithm 1), this key observation is used to maintain $H_t$ in line 4.

**Computing $X_t$ in line 5.** Now that $H_t$ is known at every round $t$, we can replace $P_{\mathcal{G}}(X_{t-1}^{-1}+g_t g_t^T/\lambda_t)$ in (9) with $H_t$ as:

$$X_t = \underset{X \in S_n(\mathcal{G})^{++}}{\arg\min} \; -\log\det(X) + \mathrm{Tr}(XH_t). \tag{11}$$

For an arbitrary graph $\mathcal{G}$, solving (11) might be difficult. Theorems 3.1 and 3.2 show *embarrassingly parallelizable* explicit solutions to the subproblem (11) for tridiagonal and banded sparsity patterns.

**Theorem 3.1** (Explicit solution of (11) for tridiagonal structures/chain graph). *Let the sparsity structure $\mathcal{G}$ be a chain with edges $E_{\mathcal{G}} = \{(i,j) : |i-j| \leq 1, 1 \leq i, j \leq n\}$. Also, let $H \in S_n(\mathcal{G})$ be such that any submatrix of $H$ corresponding to a complete subgraph of $\mathcal{G}$ is positive definite, then the solution of (11) is given by $\hat{X} = LDL^T$, where the unit lower bidiagonal matrix $L$ and diagonal matrix $D$ have the following non-zero entries:*

$$L_{jj} = 1, \; L_{j+1j} = -\frac{H_{j+1j}}{H_{j+1j+1}}, \; D_{jj}^{-1} = H_{jj} - \frac{H_{j+1j}^2}{H_{j+1j+1}}, \quad j \leq n-1 \; \& \; D_{nn}^{-1} = H_{nn} \tag{12}$$

Computing this explicit solution involves conducting paralellizable operations on $2 \times 2$ principle submatrices (highlighted in red) of the tridiagonal matrix $H$ to find the $\hat{X}$ as shown in the following $3 \times 3$ example:

$$H = \begin{pmatrix} \boxed{\begin{matrix} H_{11} & H_{12} \\ H_{21} & H_{22} \end{matrix}} & 0 \\ H_{21} & H_{22} & H_{23} \\ 0 & H_{32} & H_{33} \end{pmatrix} = \begin{pmatrix} \tilde{H}_{11} & \tilde{H}_{12} & 0 \\ \tilde{H}_{21} & \tilde{H}_{22} & \tilde{H}_{23} \\ 0 & \tilde{H}_{32} & \tilde{H}_{33} \end{pmatrix} + \begin{pmatrix} g_1^2 & g_1 g_2 & 0 \\ g_1 g_2 & g_2^2 & g_2 g_3 \\ 0 & g_2 g_3 & g_3^2 \end{pmatrix} \tag{13}$$

$$\rightarrow \hat{X} = \begin{pmatrix} \boxed{1} & 0 & 0 \\ \frac{H_{21}}{H_{22}} & 1 & 0 \\ 0 & -\frac{H_{32}}{H_{33}} & 1 \end{pmatrix} \begin{pmatrix} \boxed{H_{11} - \frac{H_{21}^2}{H_{22}}} & 0 & 0 \\ 0 & H_{22} - \frac{H_{23}^2}{H_{33}} & 0 \\ 0 & 0 & H_{33} \end{pmatrix} \begin{pmatrix} \boxed{1 \quad -\frac{H_{21}}{H_{22}}} & 0 \\ 0 & 1 & -\frac{H_{32}}{H_{33}} \\ 0 & 0 & 1 \end{pmatrix}$$

Conducting these operations take $\mathcal{O}(n)$ time and memory complexity, and similarly the descent direction can be found sequentially by $X_t g_t = L(D(L^T g_t))$, which can take $\mathcal{O}(n)$ time complexity, due to unit lower bidiagonal structure of $L$, furthermore, these operations can be easily parallelized. We also generalize the explicit solution to banded sparsity structures with band size $b$.

**Theorem 3.2** (Explicit solution of (11) for banded structures). *Let the sparsity pattern $\mathcal{G}$ be a banded matrix of band size $b$, i.e. $E_{\mathcal{G}} = \{(i,j) : |i-j| \leq b, 1 \leq i, j \leq n\}$. For every vertex $j$, let $I_j = \{j+1, \ldots, j+b\}$. Then $X_t = LDL^T$ is the solution of (11) with nonzero entries of $L$ and $D$ defined as follows :*

$$L_{jj} = 1, \; L_{I_j j} = -H_{I_j I_j}^{-1} H_{I_j j}, \quad D_{jj}^{-1} = (H_{jj} - H_{I_j j}^T H_{I_j I_j}^{-1} H_{I_j j}), \; 1 \leq j \leq n. \tag{14}$$

*where, $H \in S_n(\mathcal{G})$ any submatrix of $H$ corresponding to a complete subgraph of $\mathcal{G}$ is positive definite.*

Note that Theorem 3.1 is a special case of Theorem 3.2 when $b$ is set to 1, and the proof for Theorem 3.2 is given in Appendix A.1. Computing the above solution requires solving $n$ linear systems of size $b$ (which is small) as shown in Algorithm 2, and takes $\mathcal{O}((n-b+1)b^3)$ flops. Since $b \ll n$, the number of flops is $\mathcal{O}(n)$.

### 3.3 Regret bound analysis of SONew

The following theorem establishes optimal regret guarantee [26] for SONew in the online convex optimization framework mentioned in Section 3.1.

**Theorem 3.3.** *When $\mathcal{G}$ = tridiagonal/chain graph as defined in Theorem 3.1, then setting $\epsilon = \hat{\epsilon}G_\infty\sqrt{T}$, $\lambda_t = G_\infty\sqrt{t}$ and $\eta_t = \frac{D_2}{\hat{\epsilon}\sqrt{n}}$ in Algorithm 1, where $\|w_t - w^*\|_2 \leq D_2$, $\|g_t\|_\infty \leq G_\infty$ incurs a regret $R_T = \mathcal{O}(\sqrt{n}G_\infty D_2\sqrt{T})$.*

The proof sketch involves deriving an explicit expression for entries of $X_t^{-1}$ in Lemma A.2, to upper bound the term $(w_t - w^*)^T(X_t^{-1} - X_{t-1}^{-1})(w_t - w^*)$ in regret upper bound (3). Upper bounding $\frac{\eta}{2}\sum_{t=1}^T g_t^T X_t g_t$ involves using the Loewner order $X_t \preceq \|X_t\|_2 I_n \preceq \|X_t\|_\infty I_n$. A detailed proof sketch and proof is given in Appendix A.2. We note here that though the regret bound presented here is for convex losses, there are connections to non-convex convergence guarantees by using OCO (online convex optimization) learners, presented in Appendix A.2.5. While our main focus is on deep neural network training, which is typically non-convex, we also conducted convex experiments in Table 9.

| Optimizer | Time complexity | Memory complexity |
|---|---|---|
| Adam | $\mathcal{O}(d_1 d_2)$ | $d_1 d_2$ |
| rfdSON(m) | $\mathcal{O}(m^2 d_1 d_2)$ | $m d_1 d_2$ |
| Shampoo | $\mathcal{O}(d_1^3 + d_2^3)$ | $(d_1^2 + d_2^2)$ |
| tridiag-SONew | $\mathcal{O}(d_1 d_2)$ | $2 d_1 d_2$ |
| band-4-SONew | $\mathcal{O}(d_1 d_2)$ | $5 d_1 d_2$ |

Table 1: Consider preconditioning a $d_1 \times d_2$ parameter matrix. Time complexity of tridiag and banded SONew inverse scale linearly with number of parameters, but, Shampoo is cubic in the dimensions of the matrix. Memory used to store second-moments of gradients by tridiag-SONew can be significantly lower than Shampoo, for e.g. if $d_1 = 4 d_2$, then Shampoo takes $> 2$ times more memory.

## 3.4 Numerical Stability of SONew

In Theorem 3.1 and Theorem 3.2, as mentioned, any submatrix of $H_t$ corresponding to a complete subgraph of $\mathcal{G}$ should be positive definite, however, in practice, due to finite precision, each entry of $H$ is inherently perturbed with an error proportional to $\mathcal{O}(\epsilon_{mach})$, where $\epsilon_{mach}$ is machine epsilon [27]. We notice in practice that the subtraction operation in $D_{jj}^{-1} = S_{jj} = H_{jj} - H_{j+1j}^2/H_{j+1j+1}$ (line 7 Algorithm 2), which has a condition number $\kappa_{sub} = |H_{jj}|/|S_{jj}|$, can be high as $S_{jj}$ can be arbitrarily low due to near singular submatrices $\begin{bmatrix} H_{ii} & H_{ii+1} \\ H_{i+1i} & H_{i+1i+1} \end{bmatrix}$. Thus small perturbation in $H$ can lead to high perturbations in the preconditioner $\hat{X}$. We formalize this notion by deriving an end-to-end componentwise condition number (pg. 135, problem 7.11 in [27]) of SPARSIFIED_INVERSE in Theorem A.10, Appendix A.3. To reduce this condition number upper bound and be robust to perturbations in $H_t$ caused by finite precision, for a tridiagonal graph $\mathcal{G}$, we can remove edges $(j, j+1)$ which correspond to low $S_{jj} < \gamma$, where $\gamma \geq 0$ denotes a tolerance parameter. We show in Theorem A.11, Appendix A.3 that this reduces the condition number upperbound of SPARSIFIED_INVERSE. Furthermore, we generalize this to banded sparsity pattern in Algorithm 3, Appendix A.3.

## 4 Related Work

Online Newton method is a second order method in online convex optimization framework with properties such as scale invariance [35] and logarithmic regrets in exp-concave and strongly convex functions [25, 26]. However, it has a time complexity of $\mathcal{O}(n^2)$, making it infeasible for large $n$. However, introduction of LogDet divergence measure in SONew allows us to set different sparsity graphs as $\mathcal{G}$ such as banded graph with band-size $b$, for which our preconditioning process is more computationally efficient with a time complexity of $\mathcal{O}(b^3(n - b + 1))$ compared to online-newton method $\mathcal{O}(n^2)$.

Shampoo [24, 5] approximates full gradient statistics matrix using Kronecker factored preconditioners to reduce the memory and time complexity from $\mathcal{O}(n^2)$ to $\mathcal{O}(d_1^2 + d_2^2)$ and $\mathcal{O}(d_1^3 + d_2^3)$ respectively. Here, $n = d_1 d_2$ denotes number of parameters for a linear layer of dimensions $d_1 \times d_2$. The time complexity of matrix inversion takes a heavy toll in Shampoo's compute time even with the Kronecker product assumption on the preconditioner, whereas, our method has a time complexity of $\mathcal{O}(b^3 d_1 d_2)$ quadratic in dimensions of the linear layer (note that $b = 1$ for tridiagonal structure).

KFAC [38], similar to Shampoo, uses Kronecker factored preconditioning, but to approximate the Fisher-information matrix. FishLeg [20] instead approximates the inverse Fisher matrix directly by expressing it in terms of the solution to an optimisation problem. Both these methods have memory and time complexity similar to Shampoo. In this work, we compare with Shampoo among the class of Kronecker factored optimizers due to its widespread testing and adoption within the community [46]. We also point the readers to Eva [52], a concurrent work aimed at devising memory efficient optimizer by maintaining rank one matrix approximation to the Kronecker factors of KFAC matrices. For completeness, we include comparison with KFAC, FishLeg, and Eva on Autoencoder benchmark.

There is prior work [35, 36] in reducing the complexity - $\mathcal{O}(n^2)$ flops of Online Newton Step (ONS) to $\mathcal{O}(n)$ flops using sketching. These ONS variants maintain a low rank approximation of $H_t$ (as in Algorithm 1) and updating it with a new gradient $g_t$ at every iteration requires conducting SVD [36]/orthonormalization [35] of a tall and thin matrix in $\mathbb{R}^{n \times r}$, where $r$ denotes the rank of approximation of $H_t$. In Section 5, we conduct large scale experiments and compare SONew against rfdSON [37] as it's more stable than Oja-SON [35].

Table 2: **float32 experiments on Autoencoder benchmark.** We observe that diag-SONew performs the best among all first order methods while taking similar time. tridiag and band-4 SONew perform significantly better than first order methods while requiring similar linear space and time. Shampoo performs best but takes $\mathcal{O}(d_1^3 + d_2^3)$ time for computing preconditioner of a linear layer of size $d_1 \times d_2$, whereas our methods take $\mathcal{O}(d_1 d_2)$ time, as mentioned in Section 3.3. rfdSON takes similar space as SONew but performs considerably worse.

| Optimizer | First Order Methods | | | | Second Order Methods | | | | |
|---|---|---|---|---|---|---|---|---|---|
| | Adagrad | RMSProp | Adam | diag-SONew | Shampoo(20) | rfdSON(1) | rfdSON(4) | tridiag-SONew | band-4-SONew |
| Train CE loss | 54.393 | 53.330 | 53.591 | 53.025 | 50.702 | 56.21 | 55.55 | 51.723 | 51.357 |
| Time(s) | 62 | 62 | 62 | 63 | 371 | 85 | 300 | 70 | 260 |

LogDet problem in equation 11 is closely related to the Maximum Determinant Matrix Completion (MDMC) [4, 49]. The MDMC problem is the dual of LogDet problem (11), and has explicit solutions for chordal graphs [4]. Thus the explicit solutions in (14) are the same as the ones proved in [4]. Also, we noticed that the tridiagonal explicit solution has been used previously in KFAC [38] in the context of a gaussian graphical model interpretation of gradients, specifically, KFAC used a block-tridiagonal preconditioner to incorporate correlation within consecutive layers.

# 5 Experimental Results

We describe our experiments on standard Autoencoder benchmark [42] trained on MNIST dataset [12], Vision Transformer [13] on Imagenet training, GraphNetwork [6, 21] on OGBG-molpcba dataset [30], and a Large Language Model [47]. For all second order optimizers, we use grafting [2], a technique used to transfer step size between optimization algorithms. Specifically, given an update $v_1$ of Optimizer-1 and $v_2$ of Optimizer-2, grafting allows to use the direction suggested by Optimizer-2 with step size suggested by Optimizer-1. The final update is given by $\frac{\|v_1\|}{\|v_2\|} \cdot v_2$. Grafting has been shown to take advantage of a tuned optimizer step size and improve performance. For SONew and rfdSON, we use Adam grafting - using Adam optimizer step size $\|v_1\|$ with SONew/rfdSON direction $v_2/\|v_2\|$. For Shampoo, we use its default RMSProp grafting. We couldn't find rfdSON official implementation, so we use our own implementation using which we reproduced the numbers on convex losses (Appendix A.4) reported in their paper [37].

## 5.1 Autoencoder benchmark

**Setup:** We use three sparsity patterns for SONew - a) diagonal sparsity, resulting in a diagonal preconditioner similar to adaptive first order methods like Adam and Adagrad; b) tridiagonal sparsity, corresponding to a chain graph; and c) banded sparsity, represented by "band-$k$" in tables and figures for band size of $k$. We compare SONew against widely used first order methods including SGD [32]), SGD with Momentum [41], Nesterov [40], Adagrad [14], Adam [33], and Rmsprop [48]. We also compare with rfdSON [37], a recently proposed memory efficient second order optimizer and with Shampoo [24], a state of the art second-order optimizer used in practice, albeit with considerable memory and time requirements. Because of space constraint, we report only the best performing first order methods while include the entire set in the appendix. As previously mentioned, rfdSON maintains a low rank approximation of the Online Newton method's statistics matrix $\sum_i g_i g_i^T$. We observed rfdSON with adam grafting always performed better than without grafting, hence report the corresponding numbers. We evaluate rfdSON with rank $m$ approximation, denoted as rfdSON($m$), which requires $(m+1) * \#params$ space when using grafting. For a fair comparison with tridiag-SONew and band-4-SONew, we test rfdSON with $m = 1$ and $m = 4$, respectively. For shampoo, computing preconditioner at every step could be infeasible, instead it is computed every $t$ steps - referred to as Shampoo($t$). Section 3.3 compares time and memory complexities of rfdSON, Shampoo, tridiag-SONew, band-4-SONew. Note that $d_1^2 + d_2^2 \geq 2d_1 d_2 \ \forall d_1, d_2$, thus memory used by tridiag-SONew is never more than Shampoo. We use a 2.72M parameters Autoencoder and each experiment is performed using one V100 GPU having 16 GB memory. Further setup details are given in Appendix A.4.

**Results:** In Table 2 we observe that among first order methods, diag-SONew performs the best while taking same amount of time. Increasing the number of edges in the sparsity graph to tridiag or banded sparsity with band size 4 enhances the performance further. Tridiag-SONew runs $5\times$ faster than Shampoo at a marginal cost to the loss - even when Shampoo updates preconditioner once every 20 steps. Using same space, rfdSON performs considerably worse than SONew. To test the numerical stability and robustness of SONew, we reduce the precision to bfloat16 and conduct similar

| (a) VIT validation error | (b) GraphNetwork validation avg. precision |
|---|---|

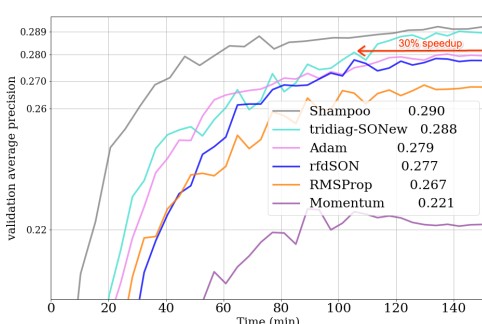

Figure 1: (a) Best validation error runs for tridiag-SONew vs Momentum, RMSProp, Adam, rfdSON, and Shampoo on (a) VIT benchmark (b) GraphNetwork benchmark. We notice that tridiag-SONew achieves same performance as Adam, the next best baseline using similar space and time, with 10% and 30% less steps/time in ViT and GraphNetwork respectively. While using the same number of steps, SONew achieves relatively 0.7% and ∼ 3.4% better validation error respectively. Shampoo doesn't fit in the 16GB memory of TPU v2 for ViT benchmark, hence we couldn't perform hyperparameter tuning on it. On GraphNetwork, compared to Shampoo, tridiag-SONew gives similar performance while being far more memory efficient (Refer Appendix A.4.2 for more details).

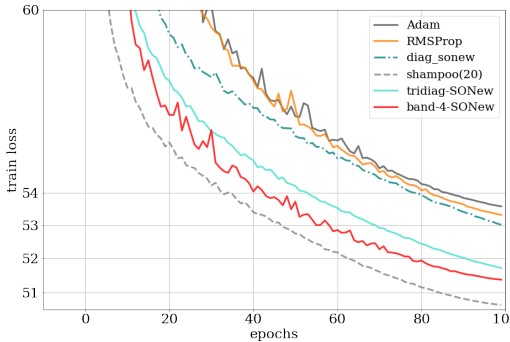

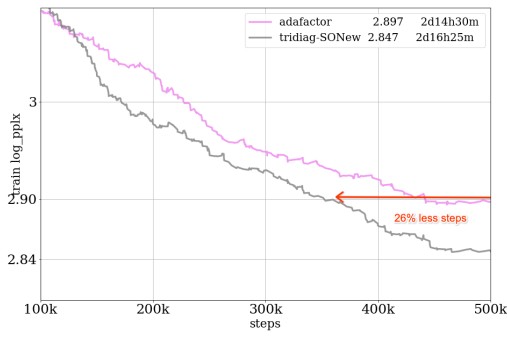

Figure 2: (a) Comparison of SONew with first-order optimizers, rfdSON, and Shampoo on Autoencoder benchmark in float32 training. We observe that SONew performs better than all first order methods, and second order methods using the same memory.

Figure 3: Comparison of SONew and Adafactor on LLM training. SONew takes 26% less steps to reach the same performance as AdaFactor. Using the same number of steps, it achieves ∼ 1.7% relative better log perplexity.

experiments in Appendix A.4.4 (Table 8 ). We notice that SONew undergoes the least degradation in performance compared to all other optimizers. We refer the reader to Appendix A.4.4 for a thorough comparison and study of bfloat16 experiments. In Figure 2 we plot the loss curves of all the baselines and SONew for float32 experiments. Moreover, in Appendix A.4.1 Table 4 we provide ablation on performance of SONew with varying batch sizes.

**Conparison with other baselines:** We further compare SONew with KFAC [38], FishLeg [20], and Eva [52] for completeness. Since these methods lack a JAX implementation, we adopted the authors' official Pytorch implementations. When we attempted to integrate their code with our Autoencoder benchmark, the results were unsatisfactory; for instance, FishLeg recorded a loss of approximately ∼ 60.0. This was notably unexpected as it underperformed Adam, a benchmark that the authors themselves compared against. Given these results and to minimize modifications to the official code, we decided to test our optimizer, SONew, directly on their provided autoencoder architecture. We present the results in Appendix A.4.4 and notice that SONew outperforms these baselines as well by a large margin.

## 5.2 VIT and GraphNetwork benchmark

**Setup:** We compare tridiag-SONew with Momentum, RMSProp, and Adam, on VIT (∼22M parameters) and GraphNetwork (∼3.5M parameters) benchmark. For each experiment, we search

over 200 hyperparameters using 4 16 GB TPUs (v2) for each run. In order to conduct a fair comparison of the running times, we executed the optimal hyperparameter configurations on 4 32GB TPUs (v4) [31]. This is because certain operations, including reshapes and transposes, are not optimized on TPUs (v2). Consequently, methods like rfdSON, Shampoo or SONew, which utilize these operations, could potentially be disadvantaged if TPUs (v2) were used, skewing the comparative results. All memory-efficient methods, including rfdSON, first-order methods, and SONew, exhibit similar runtimes, with differences of approximately $\sim 5\%$. For ViT, we evaluate their performance based on the same number of steps, as this also effectively compares wall-clock time. However, for GraphNetwork, we train Shampoo for 20% fewer steps to achieve a comparable wall-clock time.

**Results:** We plot the runs that give best validation error rate (for VIT) or validation average precision (for GraphNetwork) in Figure 1. tridiag-SONew requires $\sim 10\%$ less steps to reach the same performance as Adam for VIT, and $\sim 30\%$ less steps for GraphNetwork benchmark. Training for the same number of steps, we get $\sim 0.7\%$ better relative validation error for VIT and $\sim 3.4\%$ better relative validation average precision for GraphNetwork. On GraphNetwork benchmark, tridiag-SONew performs 1.3% relatively worse in average precision compared to Shampoo, while being $1.25\times$ faster. On VIT benchmark, Shampoo doesn't fit in a 16 GB TPU v2. Its statistics require 155M entries ($\sim 7 \times \#params$) while tridiag-SONew requires only 44M entries ($2 \times \#params$). Hence, we could not tune it. rfdSON takes same memory but slightly more time because of its SVD computations. We also notice rfdSON performs worse than Adam on both benchmarks; we leave a thorough investigation of this behavior as a future work.

We show in Appendix A.4 that corresponding to the best validation runs, tridiag-SONew optimizer's training loss is also less than that of Adam. Furthermore, from an optimization point of view we also show in Appendix A.4 that among all the 200 hyperparameter sweeps, the best training loss of tridiag-SONew is 9% relatively better on ViT and 80% relatively better on GraphNN than that of Adam. We further compare Adam and tridiag-SONew on a 248M parameter Transformer Model in Appendix A.4.4. In next subsection, we present results on a decoder only large scale Language Model.

### 5.3 Experiments on Language Models

**Setup:** Owing to SONew's scalability, we test it on a Large Language Model (LLM) [47] with 1 billion parameters. We compare SONew with AdaFactor (without factoring), a commonly used first order optimizer for training LLMs [51, 11]. AdaFactor is similar to Adam except that in addition it offers benefits like "parameter scaling", which has an effect of layerwise damping of the learning rate. We defer the reader to [45] for more details. We trained the LLM for 5B tokens with a batch size of $64k$ tokens. All experiments were performed on 16 TPU v4s. To support efficient training of large models, we implemented a sharded tridiag-SONew following model parallelism approach.

**Results:** We report the experiment in Figure 3 where we find that SONew beats Adafactor by a large margin. Specifically, SONew achieves the same log perplexity using 26% less steps. Moreover, using the same number of tokens, SONew achieves 1.7% relative better performance on train loss, leading to $1.35\times$ speedup. This shows the potential of SONew as a scalable optimizer that can be used to train large models while using second order information.

## 6 Conclusions and Future Work

In this paper we have introduced a novel Sparsified Online Newtwon (SONew) method that yields a computationally efficient sparse preconditioner that can effectively train very large DNNs. The time and memory complexity of SONew is linear in the number of parameters, unlike current Kronecker-factorization based second-order methods for training deep networks. Our experimental results show that SONew uses similar time as first order methods and achieves much better validation and training performance in various benchmarks. In the future, we plan to explore different sparsity graphs for which efficient solutions exist for the LogDet subproblem (11) and develop corresponding regret bound analyses. Some of the limitations of SONew include: 1) explicit solutions akin to Theorem 3.1 & 3.2 need not exist for all sparsity graphs $\mathcal{G}$; 2) Not all graphs allow for efficient optimizer implementation; 3) Among graphs permitting efficient optimizer implementation—like tridiagonal sparsity—the ordering of parameters remains unexplored. An alternative ordering might position closely related parameters adjacently, potentially enhancing performance; 4) Comprehensive exploration of methods to scale final updates is needed. While we employ grafting [2], other techniques, such as clipping [45, 38], merit investigation.

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

# A  Supplementary material

## A.1  Properties of LogDet subproblem

*Proof of Theorem 3.2*

The optimality condition of (11) is $P_{\mathcal{G}}(X^{-1}) = P_{\mathcal{G}}(H)$, $X \in S_n^{++}(\mathcal{G})$. Let $Z = L^{-T} D^{-1} L^{-1}$, then $P_{\mathcal{G}}(Z) = H$

$$ZL = L^{-T} D^{-1} \implies ZLe_j = L^{-T} D^{-1} e_j$$

Let $J_j = I_j \cup j$, where $I_j = \{j+1, \ldots, j+b\}$ as defined in the theorem, then select $J_j$ indices of vectors on both sides of the second equality above and selecting the $J_j$ indices :

$$\begin{bmatrix} Z_{jj} & Z_{jI_j} \\ Z_{I_jj} & Z_{J_jJ_j} \end{bmatrix} \begin{bmatrix} 1 \\ L_{I_j} \end{bmatrix} = \begin{bmatrix} 1/d_{jj} \\ 0 \end{bmatrix} \tag{15}$$

Note that $L^{-T}$ is an upper triangular matrix with ones in the diagonal hence $J_j^{th}$ block of $L^{-T} e_j$ will be $[1, 0, 0, \ldots]$. Also, since $P_{\mathcal{G}}(Z) = H$

$$\begin{bmatrix} Z_{jj} & Z_{jI_j} \\ Z_{I_jj} & Z_{J_jJ_j} \end{bmatrix} = \begin{bmatrix} H_{jj} & H_{jI_j} \\ H_{I_jj} & H_{J_jJ_j} \end{bmatrix}$$

Substituting this in the linear equation 15

$$\begin{bmatrix} H_{jj} & H_{jI_j} \\ H_{I_jj} & H_{J_jJ_j} \end{bmatrix} \begin{bmatrix} 1 \\ L_{I_j} \end{bmatrix} = \begin{bmatrix} 1/d_{jj} \\ 0 \end{bmatrix}$$

$$\begin{bmatrix} H_{jj} & H_{jI_j} \\ H_{I_jj} & H_{J_jJ_j} \end{bmatrix} \begin{bmatrix} d_{jj} \\ d_{jj} \cdot L_{I_j} \end{bmatrix} = \begin{bmatrix} 1 \\ 0 \end{bmatrix}$$

$$H_{jj} d_{jj} + d_{jj} H_{I_jj}^T L_{I_jj} = 1$$

$$H_{I_jj} d_{jj} + d_{jj} H_{I_jI_j} L_{I_jj} = 0$$

The lemma follows from solving the above equations. Note that here we used that lower triangular halves of matrices $L$ and $H$ have the same sparsity patterns, which follows from the fact that banded

graph is a chordal graph with perfect elimination order $\{1, 2, \ldots, n\}$. Furthermore, $X_t$ is positive definite, since as $(H_{jj} - H_{I_j j}^T H_{I_j I_j}^{-1} H_{I_j j})$ is a schur complement of submatrix of $H$ formed by $J_j = I_j \cup \{j\}$.

*Proof of Theorem 3.1* The proof follows trivially from Theorem 3.1, when $b$ is set to 1.

## A.2 Regret bound analysis

*Proof sketch of Theorem 3.3.* We decompose the regret into $R_T \le T_1 + T_2 + T_3$ in Lemma A.1 and individually bound the terms. Term $T_2 = \frac{1}{2\eta} \cdot \sum_{t=1}^{T-1} (w_{t+1} - w^*)^T (X_{t+1}^{-1} - X_t^{-1})(w_{t+1} - w^*)$ depends on closeness of consecutive inverses of preconditioners, $(X_{t+1}^{-1} - X_t^{-1})$, to upperbound this we first give explicit expressions of $X_t^{-1}$ for tridiagonal preconditioner in Lemma A.2 in Appendix A.2.2. This explicit expression is later used to bound each entry of $(X_{t+1}^{-1} - X_t^{-1})$ with $O(1/\sqrt{t})$ in Appendix A.2.4, this gives a $O(\sqrt{T})$ upperbound on $T_2$. To show an upperbound on $T_3 = \sum_{t=1}^{T} \frac{\eta}{2} \cdot g_t^T X_t g_t$, we individually bound $g_t^T X_t g_t$ by using a Loewner order $X_t \preceq \|X_t\|_2 I_n \preceq \|X_t\|_\infty I_n$ and show that $\|X_t\|_\infty = \mathcal{O}(1/\sqrt{T})$ and consequently $T_3 = \mathcal{O}(\sqrt{T})$.

### A.2.1 Regret bound decomposition

In this subsection we state Lemma A.1 which upper bound the regret $R_T$ using three terms $T_1$, $T_2$, $T_3$.

**Lemma A.1** ( [26] )**.** *In the OCO problem setup, if a prediction $w_t \in \mathbb{R}^n$ is made at round $t$ and is updated as $w_{t+1} := w_t - \eta X_t g_t$ using a preconditioner matrix $X_t \in S_n^{++}$*

$$R_T \le \frac{1}{2\eta} \cdot (\|w_1 - w^*\|_{X_1^{-1}}^2 - \|w_{T+1} - w^*\|_{X_T^{-1}}^2) \tag{16}$$

$$+ \frac{1}{2\eta} \cdot \sum_{t=1}^{T-1} (w_{t+1} - w^*)^T (X_{t+1}^{-1} - X_t^{-1})(w_{t+1} - w^*) \tag{17}$$

$$+ \sum_{t=1}^{T} \frac{\eta}{2} \cdot g_t^T X_t g_t \tag{18}$$

*Proof.*

$$\begin{aligned}
\|w_{t+1} - w^*\|_{X_t^{-1}}^2 &= \|w_t - \eta X_t g_t - w^*\|_{X_t^{-1}}^2 \\
&= \|w_t - w^*\|_{X_t^{-1}}^2 + \eta^2 g_t^T X_t g_t \\
&\quad - 2\eta(w_t - w^*)^T g_t \\
\implies 2\eta(w_t - w^*)^T g_t &= \|w_t - w^*\|_{X_t^{-1}}^2 - \|w_{t+1} - w^*\|_{X_t^{-1}}^2 \\
&\quad + \eta^2 g_t^T X_t g_t
\end{aligned}$$

$\square$

Using the convexity of $f_t$, $f_t(w_t) - f_t(w^*) \le (w_t - w^*)^T g_t$, where $g_t = \Delta f_t(w_t)$ and summing over $t \in [T]$

$$R_T \le \sum_{t=1}^{T} \frac{1}{2\eta} \cdot \left( \|w_t - w^*\|_{X_t^{-1}}^2 - \|w_{t+1} - w^*\|_{X_t^{-1}}^2 \right) \tag{19}$$

$$+ \frac{\eta}{2} \cdot g_t^T X_t g_t \tag{20}$$

The first summation can be decomposed as follows

$$\sum_{t=1}^{T} \left( \|w_t - w^*\|^2_{X_t^{-1}} - \|w_{t+1} - w^*\|^2_{X_t^{-1}} \right)$$

$$= \left( \|w_1 - w^*\|^2_{X_1^{-1}} - \|w_{T+1} - w^*\|^2_{X_T^{-1}} \right)$$

$$+ \sum_{t=1}^{T-1} (w_{t+1} - w^*)^T (X_{t+1}^{-1} - X_t^{-1})(w_{t+1} - w^*)$$

Substituting the above identity in the Equation (19) proves the lemma.

Let $R_T \leq T_1 + T_2 + T_3$, where

- $T_1 = \frac{1}{2\eta} \cdot (\|w_1 - w^*\|^2_{X_1^{-1}} - \|w_{T+1} - w^*\|_{X_T^{-1}})$
- 

$$T_2 = \frac{1}{2\eta} \cdot \sum_{t=1}^{T-1} (w_{t+1} - w^*)^T (X_{t+1}^{-1} - X_t^{-1})(w_{t+1} - w^*) \tag{21}$$

- $T_3 = \sum_{t=1}^{T} \frac{\eta}{2} \cdot g_t^T X_t g_t$

### A.2.2 Properties of tridiagonal preconditioner

In this subsection, we derive properties of the tridigonal preconditioner obtained from solving the LogDet subproblem (11) with $\mathcal{G}$ set to a chain graph over ordered set of vertices $\{1, \ldots, n\}$:

$$X_t = \operatorname*{arg\,min}_{X \in S_n(\mathcal{G})^{++}} -\log \det (X) + \operatorname{Tr}(X H_t) \tag{22}$$

$$= \operatorname*{arg\,min}_{X \in S_n(\mathcal{G})^{++}} \mathrm{D}_{\ell d} (X, H_t^{-1}) \tag{23}$$

The second equality holds true only when $H_t$ is positive definite. Although in Algorithm 1 we maintain a sparse $H_t = H_{t-1} + P_{\mathcal{G}}(g_t g_t^T / \lambda_t)$, $H_0 = \epsilon I_n$ which is further used in (22) to find the preconditioner $X_t$, our analysis assumes the full update $H_t = H_{t-1} + g_t g_t^T / \lambda_t$, $H_0 = \epsilon I_n$ followed by preconditioner $X_t$ computation using (23). Note that the preconditioners $X_t$ generated both ways are the same, as shown in Section 3.2.

The following lemma shows that the inverse of tridiagonal preconditioners used in Algorithm 1, will restore $H_{i,j}$, when $(i, j)$ fall in the tridiagonal graph, else, the expression is related to product of $H_{i+k,i+k+1}$ corresponding to the edges in the path from node $i$ to $j$ in chain graph. This lemma will be used later in upperbounding $T_2$.

**Lemma A.2** (*Inverse of tridiagonal preconditioner*). *If* $\mathcal{G} = $ *chain/tridiagonal graph and* $\hat{X} = \arg\min_{X \in S_n(\mathcal{G})^{++}} \mathrm{D}_{\ell d} (X, H^{-1})$, *then the inverse* $\hat{X}^{-1}$ *has the following expression*

$$(\hat{X}^{-1})_{ij} = \begin{cases} H_{ij} & |i - j| \leq 1 \\ \frac{H_{ii+1} H_{i+1 i+2} \ldots H_{j-1 j}}{H_{i+1 i+1} \ldots H_{j-1 j-1}} \end{cases} \tag{24}$$

*Proof.*

$$\hat{X}^{-1} \hat{X}^{(j)} = e_j$$

Where $\hat{X}^{(j)}$ is the $j^{th}$ column of $\hat{X}$. Let $\hat{Y}$ denote the right hand side of Equation (24).

$$(\hat{Y}\hat{X})_{jj} = \hat{X}_{jj}\hat{Y}_{jj} + \hat{X}_{j-1j}\hat{Y}_{j-1j} + \hat{X}_{jj+1}\hat{Y}_{jj+1}$$

$$= \hat{X}_{jj}H_{jj} + \hat{X}_{j-1j}H_{j-1j} + \hat{X}_{jj+1}H_{jj+1}$$

$$= 1$$

The third equality is by using the following alternative form of Equation (12):

$$(\hat{X}^{(1)})_{i,j} = \begin{cases} 0, \text{ if } j - i > 1 \\ \frac{-H_{i,i+1}}{(H_{ii}H_{i+1,i+1}-H_{i+1,i+1}^2)}, \text{ if } j = i+1 \\ \frac{1}{H_{ii}}\left(1 + \sum_{j\in\text{neig}_{\mathcal{G}}(i)}\frac{H_{ij}^2}{H_{ii}H_{jj}-H_{ij}^2}\right), \text{ if } i = j \end{cases}, \tag{25}$$

where $i < j$. Similarly, the offdiagonals of $\hat{Y}\hat{X}$ can be evaluated to be zero as follows.

$$\begin{aligned}(\hat{Y}\hat{X})_{ij} &= \hat{Y}_{ij}\hat{X_j}j + \hat{Y}_{ij-1}\hat{X}_{j-1j} + \hat{Y}_{ij+1}\hat{X}_{j+1j} \\ &= \hat{Y}_{ij}\hat{X}_{jj} + \hat{Y}_{ij}\frac{H_{j-1j-1}}{H_{j-1j}} + \hat{Y}_{ij}\frac{H_{jj+1}}{H_{jj}}\hat{X}_{j+1j} \\ &= 0\end{aligned}$$

$\square$

**Lemma A.3.** *Let* $y \in \mathbb{R}^n$,
$\beta = \max_t \max_{i\in[n-1]} |(H_t)_{ii+1}| / \sqrt{(H_t)_{ii}(H_t)_{i+1i+1}} < 1$, *then*

$$y^T X_t^{-1} y \leq \|y\|_2^2 \|\text{diag}(H_t)\|_2 \left(\frac{1+\beta}{1-\beta}\right),$$

*where* $X_t$ *is defined as in Lemma A.2.*

*Proof.* Let $\tilde{X}_t^{-1} = \text{diag}(H_t)^{-1/2}\hat{X}_t \text{diag}(H_t)^{-1/2}$

$$y^T X_t^{-1} y \leq \left\|\text{diag}(H_t)^{1/2}y\right\|_2^2 \left\|\tilde{X}_t^{-1}\right\|_2 \tag{26}$$

Using the identity of spectral radius $\rho(X) \leq \|X\|_\infty$ and since $\tilde{X}$ is positive definite, $\left\|\tilde{X}_t^{-1}\right\|_2 \leq \|\tilde{X}_t^{-1}\|_\infty$

$$\begin{aligned}\left\|\tilde{X}_t^{-1}\right\|_2 &\leq \max_i \left\{\sum_j \left|(\tilde{X}_t^{-1})_{ij}\right|\right\} \\ &\leq 1 + 2(\beta + \beta^2 + \ldots) \\ &\leq \frac{1+\beta}{1-\beta}\end{aligned}$$

The second inequality is using Lemma A.2. Substituting this in Equation (26) will give the lemma. $\square$

### A.2.3 Upperbounding Regret

The following Lemma is used in upperbounding both $T_1$ and $T_3$. In next subsection, we'll upper bound $T_2$ as well.

**Lemma A.4.** *Let* $\beta = \max_{t\in[T]} \max_{i\in[n-1]} |(H_t)_{ii+1}| / \sqrt{(H_t)_{ii}(H_t)_{i+1i+1}}$, *then*

$$1/(1-\beta) \leq 8/\hat{\epsilon}^2,$$

*where,* $\hat{\epsilon}$ *is a constant in parameter* $\epsilon = \hat{\epsilon}G_\infty\sqrt{T}$ *and consequently used in initializing* $H_0 = \epsilon I_n$ *in line 1 in Algorithm 1,*

*Proof.*

$$1/(1-\beta) = \max_t \max_{i\in[n-1]} \frac{1}{1-\left|(\hat{H}_t)_{ii+1}\right|} \tag{27}$$

$$= \max_t \max_{i\in[n-1]} \frac{1+\left|(\hat{H}_t)_{ii+1}\right|}{1-(\hat{H}_t)_{ii+1}^2} \qquad \left(\text{where } (\hat{H}_t)_{ii+1} = \frac{(H_t)_{ii+1}}{\sqrt{(H_t)_{ii}(H_t)_{i+1i+1}}}\right)$$

$$\leq \max_t \max_{i\in[n-1]} \frac{2(H_t)_{ii}(H_t)_{i+1i+1}}{(H_t)_{ii}(H_t)_{i+1i+1}-(H_t)_{ii+1}^2} \qquad \left(\text{since } |(H_t)_{ii+1}| \leq \sqrt{(H_t)_{ii}(H_t)_{i+1i+1}}\right)$$

$$\leq \max_t \max_{i\in[n-1]} \frac{2(H_t)_{ii}(H_t)_{i+1i+1}}{\det\left(\begin{bmatrix}(H_t)_{ii} & (H_t)_{ii+1}\\(H_t)_{i+1i} & (H_t)_{i+1i+1}\end{bmatrix}\right)} \tag{28}$$

Note that $\begin{bmatrix}(H_t)_{ii} & (H_t)_{ii+1}\\(H_t)_{i+1i} & (H_t)_{i+1i+1}\end{bmatrix} \succeq \epsilon\begin{bmatrix}1 & 0\\0 & 1\end{bmatrix}$ (using line 1 in Algorithm 1), thus $\det\left(\begin{bmatrix}(H_t)_{ii} & (H_t)_{ii+1}\\(H_t)_{i+1i} & (H_t)_{i+1i+1}\end{bmatrix}\right) \geq \det\left(\epsilon\begin{bmatrix}1 & 0\\0 & 1\end{bmatrix}\right) = \epsilon^2$. The numerator last inequality can be upperbounded by bounding $(H_t)_{ii}$ individually as follows:

$$(H_t)_{ii} = \sum_{s=1}^{t}(g_s)_i^2/\lambda_s$$

$$= \sum_{s=1}^{t}(g_s)_i^2/\lambda_s$$

$$= \sum_{s=1}^{t}(g_s)_i^2/(G_\infty\sqrt{s})$$

$$\leq \sum_{s=1}^{t}G_\infty^2/(G_\infty\sqrt{s})$$

$$\leq \sum_{s=1}^{t}\frac{G_\infty}{\sqrt{s}}$$

$$\leq 2G_\infty\sqrt{t} \tag{29}$$

Substituting the above in (28) gives

$$1/(1-\beta) \leq \max_t \frac{8G_\infty^2 t}{\hat{\epsilon}^2 G_\infty^2 T}$$

$$\leq \frac{8}{\hat{\epsilon}^2}$$

$\square$

**Lemma A.5** (*Upperbound of $T_1$*)**.**

$$T_1 \leq \frac{16D_2^2 G_\infty\sqrt{T}}{\hat{\epsilon}^2\eta}, \tag{30}$$

*where $D_2 = \max_{t\in[T]}\|w_t-w^*\|_2$ and $G_\infty = \max_t\|g_t\|_\infty$*

*Proof.* Since $X_T$ is positive definite

$$
\begin{aligned}
T_1 &\leq \frac{\|w_1 - w^*\|^2_{X_1^{-1}}}{2\eta} \\
&= \frac{(y^{(1)})^T X_1^{-1} y^{(1)}}{2\eta} && \text{(where } y^{(1)} = w_1 - w^*) \\
&\leq \frac{\|y^{(1)}\|_2^2 \|\text{diag}(H_1)\|_2}{2\eta} \cdot \frac{1+\beta}{1-\beta} && \text{(Lemma A.3)} \\
&\leq \frac{D_2^2(G_\infty^2/\lambda_1 + \epsilon)}{2\eta} \cdot \frac{1+\beta}{1-\beta} && \text{(line 4 in Algorithm 1)} \\
&\leq \frac{8D_2^2(G_\infty^2/\lambda_1 + \epsilon)}{\hat{\epsilon}^2 \eta} && \text{(Lemma A.4)} \\
&\leq \frac{8D_2^2(G_\infty + \hat{\epsilon}G_\infty\sqrt{T})}{\hat{\epsilon}^2 \eta} && \text{(Since } \lambda_t = G_\infty\sqrt{t} \text{ and } \epsilon = \hat{\epsilon}G_\infty\sqrt{T}) \\
&\leq \frac{16D_2^2 G_\infty \sqrt{T}}{\hat{\epsilon}^2 \eta} && (\hat{\epsilon} < 1)
\end{aligned}
$$

$\square$

**Lemma A.6** ($O(\sqrt{T})$ upperbound on $T_3$)**.**

$$
T_3 = \sum_{t=1}^{T} \frac{\eta}{2} \cdot g_t^T X_t g_t \leq \frac{4nG_\infty\eta}{\hat{\epsilon}^3}\sqrt{T}
$$

*where, $\|g_t\|_\infty \leq G_\infty$ and parameters $\epsilon = \hat{\epsilon}G_\infty\sqrt{T}$, $\lambda_t = G_\infty\sqrt{t}$ in Algorithm 1.*

*Proof.* Using Theorem 3.1, nonzero entries of $X_t$ can be written as follows:

$$
(X_t)_{ii} = \frac{1}{H_{ii}} \left( 1 + \sum_{(i,j)\in E_\mathcal{G}} \frac{H_{ij}^2}{H_{ii}H_{jj} - H_{ij}^2} \right)
$$

$$
(X_t)_{ii+1} = -\frac{H_{ii+1}}{H_{ii}H_{i+1i+1} - H_{ii+1}^2}
$$

where, $E_\mathcal{G}$ denote the set of edges of the chain graph $\mathcal{G}$ in Theorem 3.1. Also, for brevity, the subscript is dropped for $H_t$. Let $\hat{X}_t = \sqrt{\text{diag}(H)} X_t \sqrt{\text{diag}(H)}$, then $\hat{X}_t$ can be written as

$$
(\hat{X}_t)_{ii} = \left( 1 + \sum_{(i,j)\in E_\mathcal{G}} \frac{\hat{H}_{ij}^2}{1 - \hat{H}_{ij}^2} \right),
$$

$$
(\hat{X}_t)_{ii+1} = -\frac{\hat{H}_{ii+1}}{1 - \hat{H}_{ii+1}^2},
$$

where, $\hat{H}_{ij} = H_{ij}/\sqrt{H_{ii}H_{jj}}$. Note that $\hat{X}_t \preceq \|\hat{X}_t\|_2 I_n \preceq \|\hat{X}_t\|_\infty I_n$, using $\max\{|\lambda_1(\hat{X}_t)|, \ldots, |\lambda_n(\hat{X}_t)|\} \leq \|\hat{X}_t\|_\infty$ (property of spectral radius). So we upperbound $\|\hat{X}_t\|_\infty = \max_{i\in[n]}\{|(\hat{X}_t)_{11}| + |(\hat{X}_t)_{12}|, \ldots, |(\hat{X}_t)_{ii-1}| + |(\hat{X}_t)_{ii}| + |(\hat{X}_t)_{ii+1}|, \ldots, |(\hat{X}_t)_{nn}| + |(\hat{X}_t)_{nn-1}|\}$ next. Individual terms $|(\hat{X}_t)_{ii-1}| + |(\hat{X}_t)_{ii}| + |(\hat{X}_t)_{ii+1}|$ can be written as follows:

$$\sum_{(i,j)\in E_{\mathcal{G}}} |(\hat{X}_t)_{ij}| = 1 + \sum_{(i,j)\in E_{\mathcal{G}}} \frac{\hat{H}_{ij}^2}{1-\hat{H}_{ij}^2} + \frac{|\hat{H}_{ij}|}{1-\hat{H}_{ij}^2}$$

$$= 1 + \sum_{(i,j)\in E_{\mathcal{G}}} \frac{|\hat{H}_{ij}|}{1-|\hat{H}_{ij}|}$$

$$\leq 2 \max_{i\in[n-1]} \frac{1}{1-|\hat{H}_{ii+1}|}$$

The last inequality is because $|\hat{H}_{ij}| \leq 1$. Thus, $\|\hat{X}_t\|_\infty \leq 2\max_{i\in[n-1]} \frac{1}{1-|\hat{H}_{ii+1}|}$. Now

$$g_t^T X_t g_t \leq g_t^T \operatorname{diag}(H_t)^{-1/2} \hat{X}_t \operatorname{diag}(H_t)^{-1/2} g_t$$

$$\leq \|\hat{X}_t\|_\infty \|\operatorname{diag}(H_t)^{-1/2} g_t\|_2^2 \qquad\qquad \left(\left\|\hat{X}_t\right\|_2 \leq \left\|\hat{X}_t\right\|_\infty\right)$$

$$\leq 2 \max_{i\in[n-1]} \frac{1}{1-|\hat{H}_{ii+1}|} g_t^T \operatorname{diag}(H_t)^{-1} g_t.$$

Using $\operatorname{diag}(H_t) \succeq \epsilon I_n$ (step 1 in Algorithm 1), where $\epsilon = \hat{\epsilon} G_\infty \sqrt{T}$ as set in Lemma A.8, gives

$$g_t^T X_t g_t \leq 2 \max_{i\in[n-1]} \frac{1}{1-|\hat{H}_{ii+1}|} \frac{\|g_t\|_2^2}{\hat{\epsilon} G_\infty \sqrt{T}}$$

$$\leq 2 \max_{i\in[n-1]} \frac{nG_\infty}{\hat{\epsilon}(1-|\hat{H}_{ii+1}|)\sqrt{T}}$$

$$\leq \frac{2nG_\infty}{\hat{\epsilon}(1-\beta)\sqrt{T}} \qquad\qquad \left(\text{where } \beta = \max_{t\in[T]} \max_{i\in[n-1]} \left|(\hat{H}_t)_{ii+1}\right|\right)$$

Summing up over $t$ gives

$$\sum_t \frac{\eta}{2} g_t^T X_t g_t \leq \sum_t \frac{16nG_\infty \eta}{\hat{\epsilon}^3 \sqrt{T}} \qquad\qquad \text{(Using Lemma A.4)}$$

$$\leq \frac{16nG_\infty \eta}{\hat{\epsilon}^3} \sqrt{T}$$

$\square$

### A.2.4  $\mathcal{O}(\sqrt{T})$ Regret

In this section we derive a regret upper bound with a $\mathcal{O}(T^{1/2})$ growth. For this, we upper bound $T_2$ as well in this section. In (21), $T_2 = \sum_{t=2}^T (w_t - w^*)^T (X_t^{-1} - X_{t-1}^{-1})(w_t - w^*)$ can be upper bounded to a $\mathcal{O}(T^{1/2})$ by upperbounding entries of $X_t^{-1} - X_{t-1}^{-1}$ individually. The following lemmas constructs a telescoping argument to bound $\left|(X_t^{-1} - X_{t-1}^{-1})_{i,j}\right|$.

**Lemma A.7.** *Let* $H, \tilde{H} \in S_n^{++}$, *such that* $\tilde{H} = H + gg^T/\lambda$, *where* $g \in \mathbb{R}^n$, *then*

$$\frac{\tilde{H}_{ij}}{\sqrt{\tilde{H}_{ii}\tilde{H}_{jj}}} - \frac{H_{ij}}{\sqrt{H_{ii}H_{jj}}}$$

$$= \frac{g_i g_j}{\lambda\sqrt{\tilde{H}_{ii}\tilde{H}_{jj}}} + \frac{H_{ij}}{\sqrt{H_{ii}H_{jj}}} \left(\sqrt{\frac{H_{ii}H_{jj}}{\tilde{H}_{ii}\tilde{H}_{jj}}} - 1\right) = \theta_{ij}$$

*Proof.*

$$\frac{\tilde{H}_{ij}}{\sqrt{\tilde{H}_{ii}\tilde{H}_{jj}}} - \frac{H_{ij}}{\sqrt{H_{ii}H_{jj}}}$$

$$= \frac{1}{\sqrt{H_{ii}H_{jj}}}\left(\tilde{H}_{ij}\frac{\sqrt{H_{ii}H_{jj}}}{\sqrt{\tilde{H}_{ii}\tilde{H}_{jj}}} - H_{ij}\right)$$

$$= \frac{1}{\sqrt{H_{ii}H_{jj}}}\left(g_i g_j \frac{\sqrt{H_{ii}H_{jj}}}{\sqrt{\tilde{H}_{ii}\tilde{H}_{jj}}} + H_{ij}\left(\frac{\sqrt{H_{ii}H_{jj}}}{\sqrt{\tilde{H}_{ii}\tilde{H}_{jj}}} - 1\right)\right)$$

□

The following Lemma bounds the change in the inverse of preconditioner $Y^{-1}$, when there is a rank one perturbation to $H \succ 0$ in following LogDet problem (11) :

$$Y = \underset{X \in S_n(\mathcal{G})^{++}}{\arg\min} \ -\log\det(X) + \mathrm{Tr}(XH)$$

$$= \underset{X \in S_n(\mathcal{G})^{++}}{\arg\min} \ \mathrm{D}_{\ell d}(X, H)$$

**Lemma A.8** (*Rank 1 perturbation of LogDet problem (11)*)**.** *Let $H, \tilde{H} \in S_n^{++}$, such that $\tilde{H} = H + gg^T/\lambda$, where $g \in \mathbb{R}^n$. Also, $\tilde{Y} = \arg\min_{X \in S_n(\mathcal{G})^{++}} \mathrm{D}_{\ell d}(X, \tilde{H})$ and $Y = \arg\min_{X \in S_n(\mathcal{G})^{++}} \mathrm{D}_{\ell d}(X, H)$, where $\mathcal{G}$ is a chain graph, then*

$$\left|(\tilde{Y}^{-1} - Y^{-1})_{ii+k}\right| \le G_\infty^2 \kappa(k\beta + k + 2)\beta^{k-1}/\lambda,$$

*where $i, i+k \le n$, $G_\infty = \|g\|_\infty$ and $\max_{i,j}|H_{ij}|/\sqrt{H_{ii}H_{jj}} \le \beta < 1$. Let $\kappa(\mathrm{diag}(H)) :=$ condition number of the diagonal part of H, then $\kappa := \max(\kappa(\mathrm{diag}(H)), \kappa(\mathrm{diag}(\tilde{H})))$.*

*Proof.* Using Lemma A.2 will give the following:

$$\left|(\tilde{Y}^{-1} - Y^{-1})_{ii+k}\right|$$

$$= \left|\frac{\tilde{H}_{ii+1}\ldots\tilde{H}_{i+k-1i+k}}{\tilde{H}_{i+1i+1}\ldots\tilde{H}_{i+k-1i+k-1}} - \frac{H_{ii+1}\ldots H_{i+k-1i+k}}{H_{i+1i+1}\ldots H_{i+k-1i+k-1}}\right|$$

$$= \left|\sqrt{\tilde{H}_{ii}}\tilde{N}_{ii+1}\ldots\tilde{N}_{i+k-1i+k}\sqrt{\tilde{H}_{i+ki+k}}\right.$$

$$\left. - \sqrt{H_{ii}}N_{ii+1}\ldots N_{i+k-1i+k}\sqrt{H_{i+ki+k}}\right|$$

$$= \sqrt{\tilde{H}_{ii}\tilde{H}_{i+ki+k}}\left|\tilde{N}_{ii+1}\ldots\tilde{N}_{i+k-1i+k} - N_{ii+1}\ldots N_{i+k-1i+k}\sqrt{H_{ii}H_{i+ki+k}/\tilde{H}_{ii}\tilde{H}_{i+ki+k}}\right|$$

where $N_{ij} = H_{ij}/\sqrt{H_{ii}H_{jj}} < 1$ (Since determinants of 2x2 submatrices of H are positive). Expanding $\tilde{N}_{ii+1} = N_{ii+1} + \theta_{ii+1}$ (from Lemma A.7), subsequently $\tilde{N}_{ii+2} = N_{ii+2} + \theta_{ii+2}$ and so on will give

$$\left|\tilde{N}_{ii+1}\ldots\tilde{N}_{i+k-1i+k} - N_{ii+1}\ldots N_{i+k-1i+k}\sqrt{\frac{H_{ii}H_{i+ki+k}}{\tilde{H}_{ii}\tilde{H}_{i+ki+k}}}\right| =$$

$$\left|\theta_{ii+1}\tilde{N}_{i+1i+2}\ldots\tilde{N}_{i+k-1i+k} + N_{ii+1}\left(\tilde{N}_{i+1i+2}\ldots\tilde{N}_{i+k-1i+k} - N_{i+1i+2}\ldots N_{i+k-1i+k}\sqrt{\frac{H_{ii}H_{i+ki+k}}{\tilde{H}_{ii}\tilde{H}_{i+ki+k}}}\right)\right|$$

$$= |\theta_{ii+1}\tilde{N}_{i+1i+2}\ldots\tilde{N}_{i+k-1i+k} + N_{ii+1}\theta_{i+1i+2}\tilde{N}_{ii+3}\ldots\tilde{N}_{i+k-1i+k} + \cdots + N_{ii+1}\ldots N_{ii+k-1}\theta_{i+k-1i+k}$$

$$+ N_{ii+1}\ldots N_{ii+k}\left(1 - \sqrt{\frac{H_{ii}H_{i+ki+k}}{\tilde{H}_{ii}\tilde{H}_{i+ki+k}}}\right)|$$

$$\leq (\sum_{l=0}^{k-1}|\theta_{i+li+l+1}|)\beta^{k-1} + \beta^{k-1}\left|1 - \sqrt{\frac{H_{ii}H_{i+ki+k}}{\tilde{H}_{ii}\tilde{H}_{i+ki+k}}}\right|,$$

$$\implies \left|(\tilde{Y}^{-1} - Y^{-1})_{ii+k}\right| \leq \sqrt{\tilde{H}_{ii}\tilde{H}_{i+ki+k}}\cdot\left((\sum_{l=0}^{k-1}|\theta_{i+li+l+1}|)\beta^{k-1} + \beta^{k-1}\left|1 - \sqrt{\frac{H_{ii}H_{i+ki+k}}{\tilde{H}_{ii}\tilde{H}_{i+ki+k}}}\right|\right)$$

where $\max_{i,j}|N_{i,j}|$, $\max_{i,j}|\tilde{N}_{i,j}| \leq \beta < 1$. Expanding $\theta_{i+li+l+1}$ from Lemma A.7 in the term $|\theta_{i+li+l+1}|\sqrt{\tilde{H}_{ii}\tilde{H}_{i+ki+k}}$ will give:

$$|\theta_{i+li+l+1}|\sqrt{\tilde{H}_{ii}\tilde{H}_{i+ki+k}}$$

$$= \left|\sqrt{\tilde{H}_{ii}\tilde{H}_{i+ki+k}}\frac{g_{i+l}g_{i+l+1}}{\lambda\sqrt{\tilde{H}_{i+li+l}\tilde{H}_{i+l+1i+l+1}}} + \sqrt{\tilde{H}_{ii}\tilde{H}_{i+ki+k}}N_{i+li+l+1}\left(\sqrt{\frac{H_{i+li+l}H_{i+l+1i+l+1}}{\tilde{H}_{i+li+l}\tilde{H}_{i+l+1i+l+1}}} - 1\right)\right|$$

$$\leq \left|\sqrt{\tilde{H}_{ii}\tilde{H}_{i+ki+k}}\frac{g_{i+l}g_{i+l+1}}{\lambda\sqrt{\tilde{H}_{i+li+l}\tilde{H}_{i+l+1i+l+1}}}\right| + \left|\sqrt{\tilde{H}_{ii}\tilde{H}_{i+ki+k}}N_{i+li+l+1}\left(1 - \sqrt{\frac{H_{i+li+l}H_{i+l+1i+l+1}}{\tilde{H}_{i+li+l}\tilde{H}_{i+l+1i+l+1}}}\right)\right|$$

Since $H_{i+li+l}H_{i+l+1i+l+1} \leq \tilde{H}_{i+li+l}\tilde{H}_{i+l+1i+l+1}$,

$$1 - \sqrt{\frac{H_{i+li+l}H_{i+l+1i+l+1}}{\tilde{H}_{i+li+l}\tilde{H}_{i+l+1i+l+1}}} \leq \max\left(1 - \frac{H_{i+li+l}}{\tilde{H}_{i+li+l}}, 1 - \frac{H_{i+l+1i+l+1}}{\tilde{H}_{i+l+1i+l+1}}\right)$$

$$\leq \max\left(\frac{g_{i+l}^2}{\lambda\tilde{H}_{i+li+l}}, \frac{g_{i+l+1}^2}{\lambda\tilde{H}_{i+l+1i+l+1}}\right)$$

Using the above, $H_{i,i}/H_{j,j} \leq \kappa$, and $|g_i| \leq G_\infty, \forall i, j \in [n]$, gives

$$\sqrt{\tilde{H}_{ii}\tilde{H}_{i+ki+k}}|\theta_{i+li+l+1}| \leq G_\infty^2\kappa/\lambda + \beta G_\infty^2\kappa/\lambda$$

$$\leq G_\infty^2\kappa(1 + \beta)/\lambda$$

Thus the following part of $\left|\left(\tilde{Y}^{-1} - Y^{-1}\right)_{ii+k}\right|$ can be upperbounded:

$$\sqrt{\tilde{H}_{ii}\tilde{H}_{i+ki+k}}\left((\sum_{l=0}^{k-1}|\theta_{i+li+l+1}|)\beta^{k-1}\right) \leq G_\infty^2\kappa(1+\beta)k\beta^{k-1}/\lambda$$

Also, $\sqrt{\tilde{H}_{ii}\tilde{H}_{i+ki+k}}\beta^{k-1}\left|1 - \sqrt{\frac{H_{ii}H_{i+ki+k}}{\tilde{H}_{ii}\tilde{H}_{i+ki+k}}}\right| \leq \beta^{k-1}\kappa G_\infty^2/\lambda$, so

$$\left|\left(\tilde{Y}^{-1} - Y^{-1}\right)_{ii+k}\right| \leq G_\infty^2\kappa(k\beta + k + 2)\beta^{k-1}/\lambda$$

$\square$

**Lemma A.9** ($\mathcal{O}(\sqrt{T})$ upper bound of $T_2$). *Given that $\kappa(\mathrm{diag}(H_t)) \leq \kappa$, $\|w_t - w^*\|_2 \leq D_2$, $\max_{i,j}|(H_t)_{ij}|/\sqrt{(H_t)_{ii}(H_t)_{jj}} \leq \beta < 1$, $\forall t \in [T]$ in Algorithm 1, then $T_2$ in Appendix A.2.1 can be bounded as follows:*

$$T_2 \leq \frac{2048\sqrt{T}}{\eta\hat{\epsilon}^5}(G_\infty D_2^2)$$

*where $\lambda_t = G_\infty\sqrt{t}$, and $\epsilon = \hat{\epsilon}G_\infty\sqrt{T}$ in Algorithm 1, and $\hat{\epsilon} \leq 1$ is a constant.*

*Proof.* Note that $T_2 = \frac{1}{2\eta} \cdot \sum_{t=1}^{T-1}(w_{t+1} - w^*)^T(X_{t+1}^{-1} - X_t^{-1})(w_{t+1} - w^*) \leq \sum_{t=1}^{T-1} D_2^2 \left\|(X_{t+1}^{-1} - X_t^{-1})\right\|_2 /(2\eta)$. Using $\|A\|_2 = \rho(A) \leq \|A\|_\infty$ for symmetric matrices $A$, we get

$$
\begin{aligned}
\left\|X_{t+1}^{-1} - X_t^{-1}\right\|_2 &\leq \|X_{t+1}^{-1} - X_t^{-1}\|_\infty \\
&= \max_i(\sum_j |(X_{t+1}^{-1} - X_t^{-1})_{ij}|) \\
&\leq 16\frac{G_\infty \kappa}{\sqrt{t}(1-\beta)^2} \qquad \text{(Lemma A.8)} \\
&\leq 1024 \cdot \frac{G_\infty \kappa}{\sqrt{t}\hat{\epsilon}^4}
\end{aligned}
$$

Now using $\kappa \leq 2/\hat{\epsilon}$ (using Equation (29) and $(H_t)_{ii} > \hat{\epsilon}$) and summing up terms in $T_2$ using the above will give the result. $\qquad\square$

Putting together $T_1$, $T_2$ and $T_3$ from Lemma A.5, Lemma A.9 and Lemma A.6 respectively, when $\epsilon$, $\lambda_t$ are defined as in Lemma A.9:

$$
T_1 \leq \frac{16D_2^2 G_\infty \sqrt{T}}{\hat{\epsilon}^2 \eta},
$$

$$
T_2 \leq \frac{2048\sqrt{T}}{\eta\hat{\epsilon}^5}(G_\infty D_2^2) \tag{31}
$$

$$
T_3 \leq \frac{4nG_\infty \eta}{\hat{\epsilon}^3}\sqrt{T} \tag{32}
$$

Setting $\eta = \frac{D_2}{\hat{\epsilon}\sqrt{n}}$

$$
R_T \leq T_1 + T_2 + T_3 \leq O(\sqrt{n}G_\infty D_2\sqrt{T})
$$

### A.2.5 Non-convex guarantees

Minimizing smooth non-convex functions $f$ is a complex yet interesting problem. In Agarwal et al. [1], this problem is reduced to an online convex optimization, where a sequence of objectives $f_t(w) = f(w) + c\|w - w_t\|_2^2$ are minimized. Using this approach Agarwal et al. [1] established convergence guarantees to reach a stationary point via regret minimization. Thus non-convex guarantees can be obtained from regret guarantees and is our main focus in the paper.

### A.3 Numerical stability

In this section we conduct perturbation analysis to derive an end-to-end componentwise condition number (pg. 135, problem 7.11 in [27]) upper bound of the tridiagonal explicit solution in Theorem 3.1. In addition to this, we devise Algorithm 3 to reduce this condition number upper bound for the tridiagonal sparsity structure, and be robust to $H_t$ which don't follow the non-degeneracy condition: any principle submatrix of $H_t$ corresponding to a complete subgraph of $\mathcal{G}$.

**Theorem A.10** (Condition number of tridiagonal LogDet subproblem (11)). *Let $H \in S_n^{++}$ be such that $H_{ii} = 1$ for $i \in [n]$. Let $\Delta H$ be a symmetric perturbation such that $\Delta H_{ii} = 0$ for $i \in [n]$, and $H + \Delta H \in S_n^{++}$. Let $P_\mathcal{G}(H)$ be the input to 11, where $\mathcal{G}$ is a chain graph, then*

$$
\kappa_\infty^{\ell d} \leq \max_{i \in [n-1]} 2/(1-\beta_i^2) = \hat{\kappa}_\infty^{\ell d}, \tag{33}
$$

*where, $\beta_i = H_{ii+1}, \kappa_\infty^{\ell d} :=$ componentwise condition number of (11) for perturbation $\Delta H$.*

The tridiagonal LogDet problem with inputs $H$ as mentioned in Theorem A.10, has high condition number when $1 - \beta_i^2 = H_{ii} - H_{ii+1}^2/H_{i+1i+1}$ are low and as a result the preconditioner $X_t$ in

SONew (Algorithm 1) has high componentwise relative errors. We develop Algorithm 3 to be robust to degenerate inputs $H$, given that $H_{ii} > 0$. It finds a subgraph $\tilde{\mathcal{G}}$ of $\mathcal{G}$ for which non-degeneracy conditions in Theorem 3.2 is satisfied and (14) is well-defined. This is done by removing edges which causes inverse $H_{I_j I_j}^{-1}$ to be singular or $(H_{jj} - H_{I_j j}^T H_{I_j I_j}^{-1} H_{I_j j})$ to be low. In the following theorem we also show that the condition number upper bound in Theorem A.10 reduces in tridiagonal case. To test the robustness of this method we conducted an ablation study in Table 5, in an Autoencoder benchmark (from Section 5) in bfloat16 where we demonstrate noticeable improvement in performance when Algorithm 3 is used.

**Theorem A.11** (Numerically stable algorithm). *Algorithm 3 finds a subgraph $\tilde{\mathcal{G}}$ of $\mathcal{G}$, such that explicit solution for $\tilde{\mathcal{G}}$ in (14) is well-defined. Furthermore, when $\mathcal{G}$ is a tridiagonal/chain graph, the component-wise condition number upper bound in (33) is reduced upon using Algorithm 3, $\hat{\kappa}_{\ell d}^{\tilde{\mathcal{G}}} < \hat{\kappa}_{\ell d}^{\mathcal{G}}$, where $\hat{\kappa}_{\ell d}^{\tilde{\mathcal{G}}}$, $\hat{\kappa}_{\ell d}^{\mathcal{G}}$ are defined as in Theorem A.10 for graphs $\tilde{\mathcal{G}}$ and $\mathcal{G}$ respectively.*

The proofs for Theorems A.10 and A.11 are given in the following subsections.

---

**Algorithm 3** Numerically stable banded LogDet solution

---
1: **Input:** $\mathcal{G}-$ tridiagonal or banded graph, $H-$ symmetric matrix in $\mathbb{R}^{n \times n}$ with sparsity structure $\mathcal{G}$ and $H_{ii} > 0$, $\gamma-$ tolerance parameter for low schur complements.
2: **Output:** Finds subgraph $\tilde{\mathcal{G}}$ of $\mathcal{G}$ without any degenerate cases from Lemma A.13 and finds preconditioner $\hat{X}$ corresponding to the subgraph
3: Let $E_i = \{(i,j) : (i,j) \in E_\mathcal{G}\}$ be edges from vertex $i$ to its neighbours in graph $\mathcal{G}$.
4: Let $V_i^+ = \{j : i < j, (i,j) \in E_\mathcal{G}\}$ and $V_i^- = \{j : i > j, (i,j) \in E_\mathcal{G}\}$, denote positive and negative neighbourhood of vertex $i$.
5: Let $K = \left\{i : H_{ii} - H_{I_i i}^T H_{I_i I_i}^{-1} H_{I_i i} \text{ is undefined or } \leq \gamma\right\}$
6: Consider a new subgraph $\tilde{\mathcal{G}}$ with edges $E_{\tilde{\mathcal{G}}} = E_\mathcal{G} \setminus (\bigcup_{i \in K} E_i \cup (V_i^+ \times V_i^-))$
7: **return** $\hat{X} := \text{SPARSIFIED\_INVERSE}(\tilde{H}_t, \tilde{\mathcal{G}})$, where $\tilde{H}_t = P_{\tilde{\mathcal{G}}}(H_t)$

---

### A.3.1 Condition number analysis

**Theorem A.12** (*Full version of Theorem A.10*). *Let $H \in S_n^{++}$ such that $H_{ii} = 1$, for $i \in [n]$ and a symmetric perturbation $\Delta H$ such that $\Delta H_{ii} = 0$, for $i \in [n]$ and $H + \Delta H \succ 0$. Let $\hat{X} = \arg\min_{X \in S_n(\mathcal{G})^{++}} D_{\ell d}\left(X, H^{-1}\right)$ and $\hat{X} + \Delta \hat{X} = \arg\min_{X \in S_n(\mathcal{G})^{++}} D_{\ell d}\left(X, (H + \Delta H)^{-1}\right)$, here $\mathcal{G} :=$ chain/tridiagonal sparsity graph and $S_n(\mathcal{G})^{++}$ denotes positive definite matrices which follows the sparsity pattern $\mathcal{G}$.*

$$\kappa_{\ell d} = \lim_{\epsilon \to 0} \sup \left\{ \frac{\left|\Delta \hat{X}_{ij}\right|}{\epsilon \left|\hat{X}_{ij}\right|} : |\Delta H_{k,l}| \leq |\epsilon H_{k,l}|, (k,l) \in E_\mathcal{G} \right\}$$

$$\leq \max_{i \in [n-1]} 1/(1 - \beta_i^2)$$

*where, $\kappa_{\ell d} :=$ condition number of the LogDet subproblem, $\kappa_2(.) :=$ condition number of a matrix in $\ell_2$ norm, $\beta_i = H_{ii+1}/\sqrt{H_{ii} H_{i+1 i+1}}$*

*Proof.* Consider the offdiagonals for which $(\hat{X} + \Delta \hat{X})_{ii+1} = -H_{ii+1}/(1 - H_{ii+1}^2) = f(H_{ii+1})$, where $f(x) = -x/(1 - x^2)$. Let $y = f(x)$, $\hat{y} = f(x + \Delta x)$ and $|\Delta x/x| \leq \epsilon$ then using Taylor series

$$\left|\frac{(\hat{y} - y)}{y}\right| = \left|\frac{x f'(x)}{f(x)}\right| \left|\frac{\Delta x}{x}\right| + O((\Delta x)^2)$$

$$\implies \lim_{\epsilon \to 0} \left|\frac{(\hat{y} - y)}{\epsilon y}\right| \leq \frac{x f'(x)}{f(x)}$$

Using the above inequality, with $x := H_{ii+1}$ and $y := \hat{X}_{ii+1}$,

$$\lim_{\epsilon \to 0} \left| \frac{\Delta \hat{X}_{ii+1}}{\epsilon \hat{X}_{ii+1}} \right| \leq \frac{1 + H_{ii+1}^2}{1 - H_{ii+1}^2} \tag{34}$$

$$\leq \frac{2}{1 - H_{ii+1}^2}$$

Let $g(x) = x^2/(1-x^2)$, let $y_1 = g(w_1)$, $y_2 = g(x_2)$, $\hat{y}_1 = g(w_1 + \Delta x)$, $\hat{y}_2 = g(x_2 + \Delta x)$. Using Taylor series

$$\left| \frac{(\hat{y}_1 - y_1)}{y_1} \right| = \left| \frac{x_1 f'(x_1)}{f(x_1)} \right| \left| \frac{\Delta x_1}{x_1} \right| + O((\Delta x_1)^2)$$

$$\left| \frac{(\hat{y}_2 - y_2)}{y_2} \right| = \left| \frac{x_2 f'(x_2)}{f(x_2)} \right| \left| \frac{\Delta x_2}{x_2} \right| + O((\Delta x_2)^2)$$

$$\implies \lim_{\epsilon \to 0} \frac{\Delta y_1 + \Delta y_2}{\epsilon(1 + y_1 + y_2)} \leq \max \left( \frac{2}{1 - x_1^2}, \frac{2}{1 - x_2^2} \right)$$

Putting $x_1 := H_{ii+1}$, $x_2 := H_{ii-1}$ and analyzing $y_1 := H_{ii+1}^2/(1 - H_{ii+1}^2)$ and $y_2 := H_{ii-1}^2/(1 - H_{ii-1}^2)$ will result in the following

$$\lim_{\epsilon \to 0} \left| \frac{\Delta \hat{X}_{ii}}{\hat{X}_{ii}} \right| \leq \max \left( \frac{2}{1 - H_{ii+1}^2}, \frac{2}{1 - H_{ii-1}^2} \right) \tag{35}$$

Since $\hat{X}_{ii} = 1 + H_{ii+1}^2/(1 - H_{ii+1}^2) + H_{ii-1}^2/(1 - H_{ii-1}^2)$. Putting together Equation (35) and Equation (34), the theorem is proved. $\qquad\square$

### A.3.2 Degenerate $H_t$

In SONew (1), the $H_t = P_{\mathcal{G}}(\sum_{s=1}^{t} g_s g_s^T / \lambda_t)$ generated in line 4 could be such that the matrix $\sum_{s=1}^{t} g_s g_s^T / \lambda_t$ need not be positive definite and so the schur complements $H_{ii} - H_{ii+1}^2/H_{i+1i+1}$ can be zero, giving an infinite condition number $\kappa_{\infty}^{\ell d}$ by Theorem A.10. The following lemma describes such cases in detail for a more general banded sparsity structure case.

**Lemma A.13** (Degenerate inputs to banded LogDet subproblem). *Let $H = P_{\mathcal{G}}(GG^T)$, when $\epsilon = 0$ in Algorithm 1, where $G \in \mathbb{R}^{n \times T}$ and let $g_{1:T}^{(i)}$ be $i^{th}$ row of $G$, which is gradients of parameter $i$ for $T$ rounds, then $H_{ij} = \left\langle g_{1:T}^{(i)}, g_{1:T}^{(j)} \right\rangle$.*

- *Case 1: For tridiagonal sparsity structure $\mathcal{G}$: if $g_{1:T}^{(j)} = g_{1:T}^{(j+1)}$, then $H_{jj} - H_{jj+1}^2/H_{j+1j+1} = 0$.*

- *Case 2: For $b > 1$ in (14): If $\text{rank}(H_{J_j J_j}) = \text{rank}(H_{I_j I_j}) = b$, then $(H_{jj} - H_{I_j j}^T H_{I_j I_j}^{-1} H_{I_j j}) = 0$ and $D_{jj} = \infty$. If $\text{rank}(H_{I_j I_j}) < b$ then the inverse $H_{I_j I_j}^{-1}$ doesn't exist and $D_{jj}$ is not well-defined.*

*Proof.* For $b = 1$, if $g_{1:T}^{(j)} = g_{1:T}^{(j+1)}$, then $H_{jj+1} = H_{jj} = H_{j+1j+1} = \left\| g_{1:T}^{(j)} \right\|_2^2$, thus $H_{jj} - H_{jj+1}^2/H_{j+1j+1} = 0$.

For $b > 1$, since $H_{I_j I_j}$, using Guttman rank additivity formula, $\text{rank}(H_{jj} - H_{jj+1}^2/H_{j+1j+1}) = \text{rank}(H_{J_j J_j}) - rank(H_{I_j I_j}) = 0$, thus $H_{jj} - H_{jj+1}^2/H_{j+1j+1} = 0$.

Furthermore, if $\text{rank}(H) \leq b$, then all $b + 1 \times b + 1$ principal submatrices of $H$ have rank $b$, thus $\forall j$, $H_{J_j J_j}$ have a rank $b$, thus $D_{jj}$ for all $j$ are undefined.

$\qquad\square$

If $GG^T = \sum_{i=1}^{T} g_i g_i$ is a singular matrix, then solution to the LogDet problem might not be well-defined as shown in Lemma A.13. For instance, Case 1 can occur when preconditioning the input layer of an image-based DNN with flattened image inputs, where $j^{th}$ and $(j + 1)^{th}$ pixel can be highly correlated throughout the dataset. Case 2 can occur in the first $b$ iterations in Algorithm 1 when the rank of submatrices $\text{rank}(H_{I_j I_j}) < b$ and $\epsilon = 0$.

Table 3: **float32 experiments on Autoencoder benchmark using different band sizes.** Band size 0 corresponds to diag-SONew and 1 corresponds to tridiag-SONew. We see the training loss getting better as we increase band size

| Band size | 0 (diag-SONew) | 1 (tridiag-SONew) | 4 | 10 |
|---|---|---|---|---|
| Train CE loss | 53.025 | 51.723 | 51.357 | 51.226 |

### A.3.3 Numerically Stable SONew proof

*Proof of Theorem A.11*

Let $I_i = \{j : i < j, (i, j) \in E_{\mathcal{G}}\}$ and $I'_i = \{j : i < j, (i, j) \in E_{\tilde{\mathcal{G}}}\}$ Let $K = \{i : H_{ii} - H^T_{I_i i} H^{-1}_{I_i I_i} H_{I_i i}$ is undefined or $0, i \in [n]\}$ denote vertices which are getting removed by the algorithm, then for the new graph $\tilde{\mathcal{G}}$, $D_{ii} = 1/H_{ii}, \forall i \in K$ since $H_{ii} > 0$.
Let $\bar{K} = \{i : H_{ii} - H^T_{I_i i} H^{-1}_{I_i I_i} H_{I_i i} > 0, i \in [n]\}$. Let for some $j \in \bar{K}$, if

$$l = \arg\min \{i : j < i, i \in K \cap I_j\},$$

denotes the nearest connected vertex higher than $j$ for which $D_{ll}$ is undefined or zero, then according to the definition $E_{\tilde{\mathcal{G}}}$ in Algorithm 3, $I'_j = \{j+1, \ldots l-1\} \subset I_j$, since $D_{jj}$ is well-defined, $H_{I_j I_j}$ is invertible, which makes it a positive definite matrix (since $H$ is PSD). Since $H_{jj} - H^T_{I_j j} H^{-1}_{I_j I_j} H_{I_j j} > 0$, using Guttman rank additivity formula $H_{J_j J_j} \succ 0$, where $J_j = I_j \cup j$. Since $H_{J'_j J'_j}$ is a submatrix of $H_{J_j J_j}$, it is positive definite and hence its schur complement $H_{jj} - H^T_{I'_j j} H^{-1}_{I'_j I'_j} H_{I'_j j} > 0$. Thus for all $j \in [n]$, the corresponding $D_{jj}$'s are well-defined in the new graph $\tilde{\mathcal{G}}$.

Note that $\kappa^{\tilde{\mathcal{G}}}_{\ell d} = \max_{i \in [n-1]} 1/(1 - \beta_i^2) < \max_{i \in \bar{K}} 1/(1 - \beta_i^2) = \kappa^{\mathcal{G}}_{\ell d}$, for tridiagonal graph, where $\beta_i = H_{ii+1}$, in the case where $H_{ii} = 1$. This is because the $\arg\max_{i \in [n-1]} 1/(1 - \beta_i^2) \in K$.

### A.4 Additional Experiments, ablations, and details

### A.4.1 Ablations

**Effect of band size in banded-SONew** Increasing band size in banded-SONew captures more correlation between parameters, hence should expectedly lead to better preconditioners. We confirm this through experiments on the Autoencoder benchmark where we take band size = 0 (diag-SONew), 1 (tridiag-SONew), 4, and 10 in Table 3.

**Effect of mini-batch size** To find the effect of mini-batch size, in Table 4, We empirically compare SONew with state of the art first-order methods such as Adam and RMSProp, and second-order method Shampoo. We see that SONew performance doesn't deteriorate much when using smaller or larger batch size. First order methods on the other hand suffer significantly. We also notice that Shampoo doesn't perform better than SONew in these regimes.

Table 4: **Comparison on Autoencoder with different batch-sizes**

| Baseline\Batch size | 100 | 1000 | 5000 | 10000 |
|---|---|---|---|---|
| RMSProp | 55.61 | 53.33 | 58.69 | 64.91 |
| Adam | 55.67 | 54.39 | 58.93 | 65.37 |
| Shampoo(20) | 53.91 | 50.70 | 53.52 | 54.90 |
| tds | 53.84 | 51.72 | 54.24 | 55.87 |
| bds-4 | 53.52 | 51.35 | 53.03 | 54.89 |

**Effect of Numerical Stability Algorithm 3** On tridiag-SONew and banded-4-SONew, we observe that using Algorithm 3 improves training loss. We present in Table 5 results where we observed significant performance improvements.

Table 5: **bfloat16 experiments on Autoencoder benchmark with and without Algorithm 3.** We observe improvement in training loss when using Algorithm 3

| Optimizer | Train CE loss - without Algorithm 3 | Train CE loss - with Algorithm 3 |
|---|---|---|
| tridiag-SONew | 53.150 | 51.936 |
| band-4-SONew | 51.950 | 51.84 |

Table 6: A rough estimate of memory requirement comparisons of different optimizers tested across benchmarks.

| Benchmark | # model parameters | K-FAC | Shampoo | FishLeg | Eva | Adam | SGD+Momentum | RMSprop | tds-SONew |
|---|---|---|---|---|---|---|---|---|---|
| Autoencoder | n=1.4M | 5.56n | 6.56n | 4.28n | n | 2n | n | n | 3n |
| GraphNetwork | n=3.5M | 8.6n | 10.6n | 4.8n | n | 2n | n | n | 3n |
| Vision Transformer | n=22M | 6.4n | 7.2n | 3.7n | n | 2n | n | n | n |
| Language Model | n=1.3B | 5.6n | 6.6n | 3.3n | n | 2n | n | n | 3n |

### A.4.2 Memory Requirements

We present a list of approximate memory requirements of different optimizers across different benchmarks in Table 6. Note that for K-FAC and Shampoo, because preconditioner is updated once only a few steps, they require storing the latest computed preconditioners as well along with the statistics, causing even higher memory overhead.

### A.4.3 Hyperparaeter search space

We provide the hyperparamter search space for experiments presented in Section 5. We search over $2k$ hyperparameters for each Autoencoder experiment using a Bayesian Optimization package. The search ranges are: first order momentum term $\beta_1 \in [1e-1, 0.999]$, second order momentum term $\beta_2 \in [1e-1, 0.999]$, learning rate $\in [1e-7, 1e-1]$, $\epsilon \in [1e-10, 1e-1]$. We give the optimal hyperparameter value for each experiment in Table 12. For VIT and GraphNetwork benchmark, we search $\beta_1, \beta_2 \in [0.1, 0.999]$, $lr \in [1e-5, 1e-1]$, $\epsilon \in [1e-9, 1e-4]$, weight decay $\in [1e-5, 1.0]$, learning rate warmup $\in [2\%, 5\%, 10\%]*$total_train_steps, dropout$\in [00, 0.1]$, label smoothing over $\{0.0, 0.1, 0.2\}$ . We use cosine learning rate schedule. Batch size was kept = 1024, and 512 for Vision Transformer, and GraphNetwork respectively. We sweep over 200 hyperparameters in the search space for all the optimizers.

For rfdSON [37], there's no $\epsilon$ hyperparameter. In addition to the remaining hyperparameters, we tune $\alpha \in \{1e-5, 1.0\}$ (plays similar role as $\epsilon$) and $\mu_t \in [1e-5, 0.1]$.

For LLM [47] benchmark, we only tune the learning rate $\in [1e-2, 1e-3, 1e-4]$ while keeping the rest of the hyperparams as constant. This is due to the high cost of running experiments hence we only tune the most important hyperparameter. For Adafactor [45], we use factored=False, decay method=adam, $\beta_1 = 0.9$, weight decay$=1e-3$, decay factor=0.99, and gradient clipping=1.0.

### A.4.4 Additional Experiments

**VIT and GraphNetwork Benchmarks**: In Figure 5 we plot the training loss curves of runs corresponding to the best validation runs in Figure 1. Furthermore, from an optimization point of view, we plot the best train loss runs in Figure 6 got by searching over 200 hyperparameters. We find that tridiag-SONew is $9\%$ and $80\%$ relatively better in ViT and GraphNetwork benchmark respectively (Figure 6), compared to Adam (the next best memory efficient baseline).

**Autoencoder float32 and bfloat16 experiments**: We provide curves of all the baselines and SONew in Figure 4(a) and the corresponding numbers in Table 7 for float32 experiments.

To test numerical stability of SONew and compare it with other algorithm in low precision regime, we also conduct bfloat16 experiments on the Autoencoder benchmark (Table 8). We notice that SONew undergoes the least degradation. Tridiagonal-sparsity SONew CE loss increases by only 0.21 absolute difference (from 51.72 in float32 (7) to 51.93), whereas Shampoo and Adam incur 0.70 loss increase. It's worthwhile to note that SONew performs better than all first order methods while taking similar time and linear memory, whereas while Shampoo performs marginally better, it is $22\times$ slower

Table 7: **float32 experiments on Autoencoder benchmark.** We observe that diag-SONew performs the best among all first order methods while taking similar time. tridiag and band-4 perform significantly better than first order methods while requiring similar linear space and time. Shampoo performs best but takes $\mathcal{O}(d_1^3 + d_2^3)$ time for computing preconditioner of a linear layer of size $d_1 \times d_2$, whereas our methods take $\mathcal{O}(d_1 d_2)$ time, as mentioned in Section 3.3. rfdSON takes similar space as SONew but performs considerably worse.

| Optimizer | First Order Methods | | | | | | |
|---|---|---|---|---|---|---|---|
| | SGD | Nesterov | Adagrad | Momentum | RMSProp | Adam | diag-SONew |
| Train CE loss | 67.654 | 59.087 | 54.393 | 58.651 | 53.330 | 53.591 | 53.025 |
| Time(s) | 62 | 102 | 62 | 67 | 62 | 62 | 63 |

| Optimizer | Second Order Methods | | | | |
|---|---|---|---|---|---|
| | Shampoo(20) | rfdSON(1) | rfdSON(4) | tridiag-SONew | band-4-SONew |
| Train CE loss | 50.702 | 53.56 | 52.97 | 51.723 | 51.357 |
| Time(s) | 371 | 85 | 300 | 70 | 260 |

Table 8: **bfloat16 experiments on Autoencoder benchmark** to test the numerical stability of SONew and robustness of Algorithm 3. We notice that diag-SONew degrades only marginally (0.26 absolute difference) compared to float32 performance. tridiag-SONew and band-4-SONew holds similar observations as well. Shampoo performs the best but has a considerable drop (0.70) in performance compared to float32 due to using matrix inverse, and is slower due to its cubic time complexity for computing preconditioners. Shampoo implementation uses 16-bit quantization to make it work in 16-bit setting, leading to further slowdown. Hence the running time in bfloat16 is even higher than in float32.

| Optimizer | First Order Methods | | | | | | |
|---|---|---|---|---|---|---|---|
| | SGD | Nesterov | Adagrad | Momentum | RMSProp | Adam | diag-SONew |
| Train CE loss | 80.454 | 72.975 | 68.854 | 70.053 | 53.743 | 54.328 | 53.29 |
| Train time(s) | 36 | 43 | 37 | 36 | 37 | 38 | 44 |

| Optimizer | Second Order Methods | | | | |
|---|---|---|---|---|---|
| | Shampoo(20) | rfdSON(1) | rfdSON(4) | tridiag-SONew | band-4-SONew |
| Train CE loss | 51.401 | 57.42 | 55.53 | 51.937 | 51.84 |
| Train time(s) | 1245 | 80 | 284 | 55 | 230 |

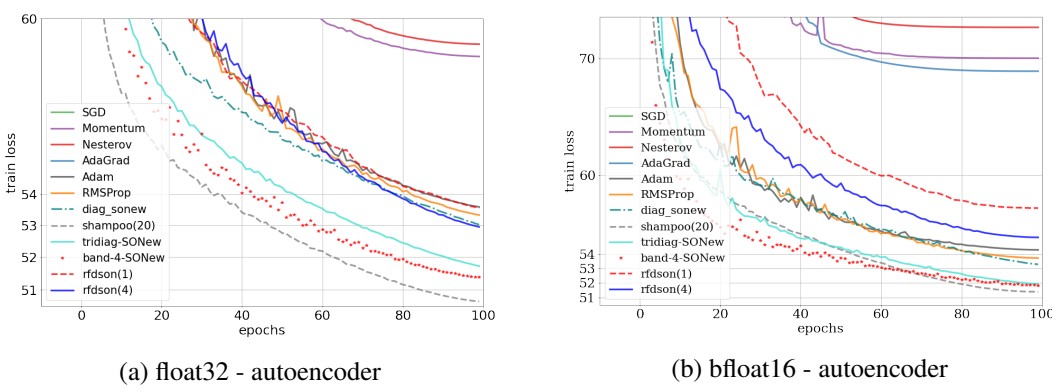

(a) float32 - autoencoder

(b) bfloat16 - autoencoder

Figure 4: Training curves of all the baselines for Autoencoder benchmar (a) float32 training (b) bfloat16 training

than tridiagonal-SONew. The corresponding loss curves are given in Figure 4(b).

**Note:** In the main paper, our reported numbers for rfdSON on Autoencoder benchmark in Table 2 for float32 experiments are erraneuous. Please consider the numbers provided in Table 7 and the corresponding curve in Figure 4(a). Note that there's no qualitiative change in the results and none of the claims made in the paper are affected. SONew is still significantly better than rfdSON. We also meticulously checked all other experiments, and they do not have any errors.

**Autoencoder on KFAC, FishLeg, Eva:** For completeness, We compare SONew against KFAC [38], FishLeg [20], and Eva [52] on Autoencoder benchmark as used in their official implementation.

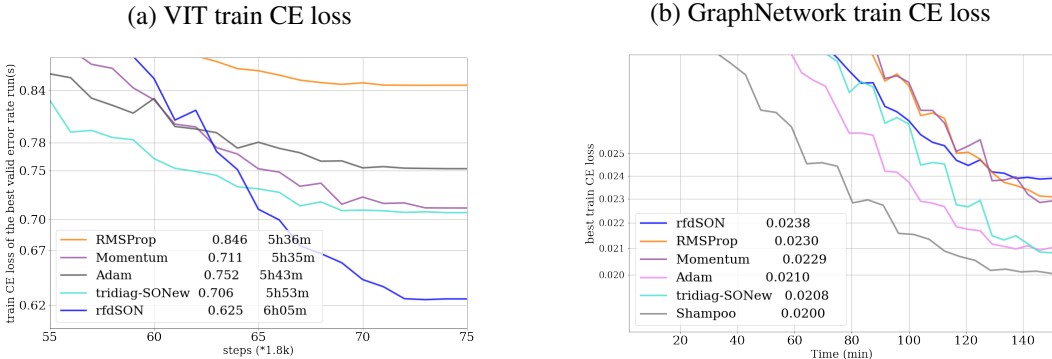

(a) VIT train CE loss

(b) GraphNetwork train CE loss

Figure 5: Train loss corresponding to the best validation runs in Figure 1 (a) VIT benchmark (b) GraphNetwork benchmark. We observe that tridiag-SONew match or perform better than Adam.

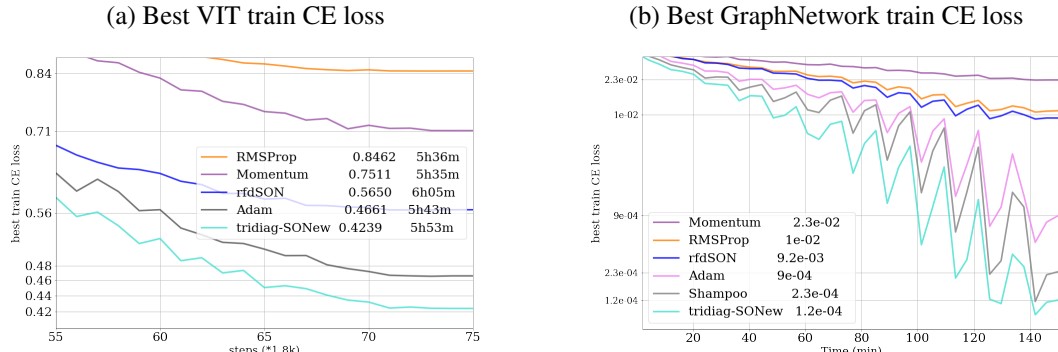

(a) Best VIT train CE loss

(b) Best GraphNetwork train CE loss

Figure 6: Best train loss achieved during hyperparam tuning. (a) VIT benchmark (b)GraphNetwork benchmark. We observe that tridiag-SONew significantly outperforms Adam, while being comparable or better than shampoo.

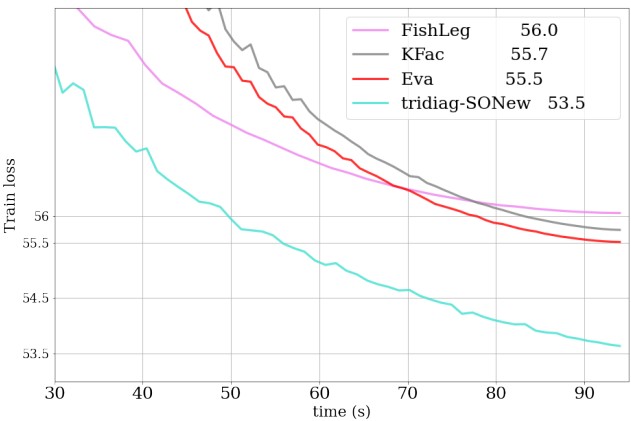

Figure 7: Autoencoder benchmark run using Pytorch on KFAC, FishLeg, Eva, and tridiag-SONew. We notice that tridiag-SONew beats all other baselines by a large margin.

The main difference is their implementation uses ReLU activation compared to Tanh that we used for all our Autoencoder experiments. As these baselines done have JAX implementation, we use their official PyTorch implementation and run tridiag-SONew in PyTorch as well. Hyperparameter search is conducted for SONew similar to as reported above, over learning rate, $\beta_1, \beta_2$, and $\epsilon$. For KFAC and Eva, rather than $\beta_2$, damping factor is tuned over $[1e-5, 10]$ (default value specified is 0.03). kl_clip is tuned as well over $[1e-5, 1.0]$. Preconditioners are updates once every 15 iterations to have same wall clock time as other baselines and SONew. For FishLeg, auxiliary learning rate is tuned $\in [1e-7, 1e-1]$ and damping $\in [1e-5, 1.0]$. All other hyperparameters are tuned similar to SONew. Eva is trained for 100 epochs, and for other methods we change number of epochs such that

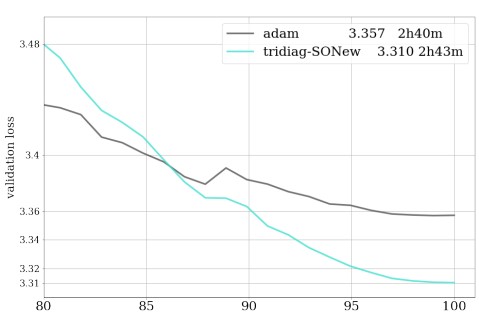
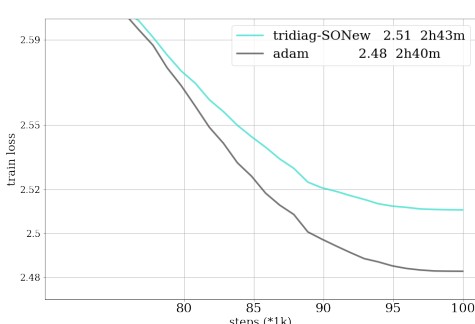

(a) Validation CE loss          (b) Train CE loss

Figure 8: We observe mixed results for 248M parameter language model benchmark. This is possibly because 48 trials were insufficient for optimal tuning. We leave thorough tuning and investigating into the above observation as future work.

each experiment takes same amount of time. Each optimizer is tuned using 600 hyperparameters. The results are in Figure 7, where notice that tridiag-SONew beats all the baselines by a large margin.

**Adam vs SONew on 248M Language Model:** We conduct an additional experiment on a 248M parameter transformer architecture [50] language model. The model is trained on WikiText103, introduced in [39]. We train the model for 3 epochs, having 8M tokens per epoch with a batch size of 8k tokens. We search over 48 hyperparameters, tuning learning rate $\in 2e-2, 1e-2, 5e-3, 1e-3$, $\beta_2 \in 0.99, 0.999$, weight decay $\in 0.0, 0.1$, and $\epsilon \in 1e-10, 1e-8, 1e-6$, while fixing $\beta_1 = 0.9$. Validation and training loss is given in Figure 8. Our observations indicate mixed results. While tds-SONew exhibits superior validation performance, Adam outperforms in training metrics. We believe that the 48 trials might have been insufficient for optimal tuning. It's conceivable that with further trials, SONew's training loss could surpass that of Adam. We leave this line of investigation for future work.

### A.4.5  Convex experiments

As our regret bound applies to convex optimization, we compare SONew to rfdSON [37], another recent memory-efficient second-order Newton method. We follow [37] for the experiment setup - each dataset is split randomly in 70%/30% train and test set. Mean squared loss is used. For tridiag-SONew, we use a total of $2 * d$ space for $d$ parameters. Hence, for fair comparison we show rfdSON with $m = 2$. Since the code isn't open sourced, we implemented it ourselves. In order to show reproducibility with respect to the reported numbers in [37], we include results with $m = 5$ as well. We see in the Table 9 that tridiag-SONew consitently matches or outperforms rfdSON across all 3 benchmarks. Each experiment was run for 20 epochs and we report the best model's performance on test set.

Table 9: **Comparison of rfdSON and tridiag-SONew in convex setting on three datasets. We optimize least square loss $\sum_t (y_t - w^T x_t)^2$ where $w$ is the learnable parameter and $(x_t, y_t)$ is the $t^{th}$ training point. Reported numbers is the accuracy on the test set.**

Table 10: (a) Dataset stats

| Dataset | # total points | dimension |
|---------|---------------|-----------|
| a9a | 32,561 | 123 |
| gisette | 6000 | 5000 |
| mnist | 11791 | 780 |

Table 11: (b) RFD-SON vs tridiag-SONew

| Dataset | RFD-SON, m=2 | RFD-SON, m=5 | tridiag-SONew |
|---------|--------------|--------------|---------------|
| a9a | 83.3 | 83.6 | 84.6 |
| gisette | 96.1 | 96.2 | 96.6 |
| mnist | 93.2 | 94.5 | 96.5 |

Table 12: **Optimal hyperparams for Autoencoder Benchmark**

Table 13: (a) float32 experiments optimal hyperparamters

| Baseline | $\beta_1$ | $\beta_2$ | $\epsilon$ | lr |
|---|---|---|---|---|
| SGD | 0.99 | 0.91 | 8.37e-9 | 1.17e-2 |
| Nesterov | 0.914 | 0.90 | 3.88e-10 | 5.74e-3 |
| Adagrad | 0.95 | 0.90 | 9.96e-7 | 1.82e-2 |
| Momentum | 0.9 | 0.99 | 1e-5 | 6.89e-3 |
| RMSProp | 0.9 | 0.9 | 1e-10 | 4.61e-4 |
| Adam | 0.9 | 0.94 | 1.65e-6 | 3.75e-3 |
| Diag-SONew | 0.88 | 0.95 | 4.63e-6 | 1.18e-3 |
| Shampoo | 0.9 | 0.95 | 9.6e-9 | 3.70e-3 |
| tridiag | 0.9 | 0.96 | 1.3e-6 | 8.60e-3 |
| band-4 | 0.88 | 0.95 | 1.5e-3 | 5.53e-3 |

Table 14: (b) bfloat16 experiments optimal hyperparamters

| Baseline | $\beta_1$ | $\beta_2$ | $\epsilon$ | lr |
|---|---|---|---|---|
| SGD | 0.96 | 0.98 | 2.80e-2 | 1.35e-2 |
| Nesterov | 0.914 | 0.945 | 8.48e-9 | 6.19e-3 |
| Adagrad | 0.95 | 0.93 | 2.44e-5 | 2.53e-2 |
| Momentum | 0.9 | 0.99 | 0.1 | 7.77e-3 |
| RMSProp | 0.9 | 0.9 | 2.53e-10 | 4.83e-4 |
| Adam | 0.9 | 0.94 | 3.03e-10 | 3.45e-3 |
| Diag-SONew | 0.9 | 0.95 | 4.07e-6 | 8.50e-3 |
| Shampoo | 0.85 | 0.806 | 6.58e-4 | 5.03e-3 |
| ztridiag | 0.83 | 0.954 | 1.78e-6 | 7.83e-3 |
| band-4 | 0.9 | 0.96 | 1.52e-6 | 4.53e-3 |

