report the numbers and the training time in the legend. We observe that tridiag significantly outperforms adam, while being comparable to shampoo.

Table 8: **Comparison of rfdSON and tridiag-SONew in convex setting on three datasets. We optimize least square loss $\sum_t (y_t - w^T x_t)^2$ where $w$ is the learnable parameter and $(x_t, y_t)$ is the $t^{th}$ training point. Reported numbers is the accuracy on the test set.**

Table 9: (a) Dataset stats

| Dataset | # total points | dimension |
|---------|----------------|-----------|
| a9a     | 32,561         | 123       |
| gisette | 6000           | 5000      |
| mnist   | 11791          | 780       |

Table 10: (b) RFD-SON vs tridiag-SONew

| Dataset | RFD-SON, m=2 | RFD-SON, m=5 | tridiag-SONew |
|---------|--------------|--------------|---------------|
| a9a     | 83.3         | 83.6         | 84.6          |
| gisette | 96.1         | 96.2         | 96.6          |
| mnist   | 93.2         | 94.5         | 96.5          |

Table 11: **Optimal hyperparams for Autoencoder Benchmark**

Table 12: (a) float32 experiments optimal hyperparamters

| Baseline | $\beta_1$ | $\beta_2$ | $\epsilon$ | lr |
|----------|-----------|-----------|------------|------|
| SGD | 0.99 | 0.91 | 8.37e-9 | 1.17e-2 |
| Nesterov | 0.914 | 0.90 | 3.88e-10 | 5.74e-3 |
| Adagrad | 0.95 | 0.90 | 9.96e-7 | 1.82e-2 |
| Momentum | 0.9 | 0.99 | 1e-5 | 6.89e-3 |
| RMSProp | 0.9 | 0.9 | 1e-10 | 4.61e-4 |
| Adam | 0.9 | 0.94 | 1.65e-6 | 3.75e-3 |
| Diag-SONew | 0.88 | 0.95 | 4.63e-6 | 1.18e-3 |
| Shampoo | 0.9 | 0.95 | 9.6e-9 | 3.70e-3 |
| tridiag | 0.9 | 0.96 | 1.3e-6 | 8.60e-3 |
| band-4 | 0.88 | 0.95 | 1.5e-3 | 5.53e-3 |

Table 13: (b) bfloat16 experiments optimal hyperparamters

| Baseline | $\beta_1$ | $\beta_2$ | $\epsilon$ | lr |
|----------|-----------|-----------|------------|------|
| SGD | 0.96 | 0.98 | 2.80e-2 | 1.35e-2 |
| Nesterov | 0.914 | 0.945 | 8.48e-9 | 6.19e-3 |
| Adagrad | 0.95 | 0.93 | 2.44e-5 | 2.53e-2 |
| Momentum | 0.9 | 0.99 | 0.1 | 7.77e-3 |
| RMSProp | 0.9 | 0.9 | 2.53e-10 | 4.83e-4 |
| Adam | 0.9 | 0.94 | 3.03e-10 | 3.45e-3 |
| Diag-SONew | 0.9 | 0.95 | 4.07e-6 | 8.50e-3 |
| Shampoo | 0.85 | 0.806 | 6.58e-4 | 5.03e-3 |
| ztridiag | 0.83 | 0.954 | 1.78e-6 | 7.83e-3 |
| band-4 | 0.9 | 0.96 | 1.52e-6 | 4.53e-3 |