# OpenReview forum: "A Computationally Efficient Sparsified Online Newton Method"
_NeurIPS.cc/2023/Conference — NeurIPS 2023 poster_

### Official Review · Reviewer_J48q · 2023-07-06

**Soundness:** 3 good
**Presentation:** 3 good
**Contribution:** 3 good
**Rating:** 7
**Confidence:** 4

**Summary:**

This paper proposes SONew -- a sparsified online Newton method, which uses the LogDet matrix divergence measure to sparsify the preconditioner. The LogDet divergence regularizes by matching the largest eigenvalues of the preconditioner matrix between iterates. It also leads to a convex optimization problem since the LogDet divergence is convex in its first argument. This leads to a simple and cost effective update rule for the preconditioner matrix. By minimizng the LogDet divergence they impose a sparsity constraint while requiring the sparse preconditioner to remain close to the full-matrix preconditioner in terms of LogDet divergence. They demonstrate the capability of their optimizer on autoencoders, vision transformers, graph neural networks, and large language models.

**Strengths:**

SONew shows faster convergence compared to first order optimizers like Adam and AdaFactor.

SONew can be applied to models sizes where Shampoo runs out of memory, while showing competitive convergence and generalization to these second-order optimizers.

When using bfloat16, SONew has the least degradation in performance compared to all other optimizers.

**Weaknesses:**

The experiments can be improved.

Table 1 only shows the final value of the train CE loss and the time to achieve it, but without looking at the training curve it is difficult to see whether SONew is converging faster or simply reaching a lower loss but slowly. What is the criteria they used to stop the training? Is the learning rate schedule tuned for each optimizer?

For the autoencoder experiments they compare with Adam, RMSProp, Shampoo(20), and diagonal, tridiagonal, and band-4 versions of SONew. For the GraphNetwork experiment they compare with Shampoo, Adam, and rfdSON. For LLMs they only compare with AdaFactor. This inconsistency makes it difficult see the effect of changing the dataset and model on all these different optimizers.

**Questions:**

How robust is SONew to perturbation in the hyperparameters? Does it require less hyperparameter tuning compared to Adam?

**Limitations:**

In the end, we need to run many hyperparameter trials to get any optimizer to achieve optimal convergence and generalization. Therefore, an optimizer that converges twice as fast, but requires more than twice the amount of hyperparameter tuning, is unlikely to be adopted by the community.

When proposing a novel optimizer, there should be more emphasis on the robustness to change in the hyperparameters, model, and dataset. This is even more critical for LLMs, where we cannot afford to run many hyperparameter trials. The criteria for a good optimizer should be this kind of robustness, and not the convergence.

---

> ### Author Rebuttal · Authors · 2023-08-10
>
> > “Table 1 only shows the final value of the train CE loss and the time to achieve it, but without looking at the training curve it is difficult to see whether SONew is converging faster or simply reaching a lower loss but slowly. What is the criteria they used to stop the training? Is the learning rate schedule tuned for each optimizer?”
>
> The time taken to finish training (for 100 epochs) is almost same for all first order methods and tridiag-SONew. Hence, a comparison wrt wall-clock time will almost be same as wrt steps. The corresponding figure is attached in Fig. 4 (a) in the Appendix and Fig (2) in the main paper. We will also make an explicit wall-clock comparison in our final draft. Moreover, we make explicit wall-clocl time comparison on OGBG benchmark in the submitted rebuttal pdf.
>
> > “For the GraphNetwork experiment they compare with Shampoo, Adam, and rfdSON. For LLMs they only compare with AdaFactor. This inconsistency makes it difficult see the effect of changing the dataset and model on all these different optimizers”
>
> Thanks for the comment. As mentioned in the Reviewer jEyd and Reviewer 3rTc, under computational constraints, we make the best calculated judgement of what baselines to compare for each benchmark. Since running Autoencoder benchmark is relatively faster, and is also commonly used to compare optimizers performance for second order methods, we make a thorough comparison on it. Moving to larger benchmarks, we choose the most commonly used first order method baseline Adam. On LLMs as well, as mentioned in the paper we choose AdaFactor, , a commonly used first order optimizer for training LLMs [46, 10].
>
> > How robust is SONew to perturbation in the hyperparameters? Does it require less hyperparameter tuning compared to Adam?”
>
> We tune all optimizers using same number of hyperparameters. In the final draft, we promise to include an ablation studying robustness with respect to different hyperparameters.
>
> > “In the end, we need to run many hyperparameter trials to get any optimizer to achieve optimal convergence and generalization. Therefore, an optimizer that converges twice as fast, but requires more than twice the amount of hyperparameter tuning, is unlikely to be adopted by the community. When proposing a novel optimizer, there should be more emphasis on the robustness to change in the hyperparameters, model, and dataset. This is even more critical for LLMs, where we cannot afford to run many hyperparameter trials. The criteria for a good optimizer should be this kind of robustness, and not the convergence”
>
> [Graft] describes grafting (which we use in SONew experiments) which the paper proposes to make the optimizers robust w.r.t hyperparameters by using grafts from tuned baselines such as Adam. To obtain the 26% improvement in LLM we use the same hyperparameter as the tuned baseline (Adafactor), however removing grafting in SONew would require us to tune the hyperparameters all over again.
>
> [Graft] Agarwal, Naman, et al. "Disentangling adaptive gradient methods from learning rates." arXiv preprint arXiv:2002.11803 (2020).

---

> > ### Comment · Reviewer_J48q · 2023-08-15
> > **Thank you for the clarification**
> >
> > The authors have addressed my concerns in their rebuttal. I am bumping up my score to an accept.

---

### Official Review · Reviewer_rRBk · 2023-07-06

**Soundness:** 3 good
**Presentation:** 3 good
**Contribution:** 2 fair
**Rating:** 7
**Confidence:** 3

**Summary:**

This paper strives to present a second-order optimization approach, called Sparsified Online Newton (SONew). SONew solves the regret minimization via introducing the LogDet matrix divergence. Using specific sparsified structures i.e. diagonal or banded sparsity, the proposed method could achieve considerable computational complexity. Finally, the authors show the effectiveness of the proposed method by performing the MNIST task with autoencoder, ImageNet task with ViT, OGBG-molpcba with GraphNetwork and language models.

**Strengths:**

***Strengths***

1. The paper is easy to follow and clearly written, although the layout of the paper is arranged too narrow.

2. The proposed method is mathematically sound and reasonable. The paper presents novel insight of online Newton methods in regards to regret minimization with LogDet divergence of the preconditioning matrix, and proposes an elegant way solve it via specific sparsified structures, and gives mathematical analysis in regards to the regret bound, which btw is O(sqrt(T)) the same with many related methods, and numerical stability.

3. Given the large computation cost and memory demand of the second-order approach, the paper shows some promising practical results in regards to the proposed methods.

**Weaknesses:**

***Weakness***

1. The authors seem to have largely changed the spacing between sections. The current spacing presented in the paper would be much more narrow than that in the latex template. Slight changes in the latex template could be accecptable, but such large changes could not be available. I could understand that the authors try to present as much content as possible. But the writing seems to be quite wordy. For example Abstract contains 24 lines, which btw is near 15 lines for other papers in common. I understand sometimes it is necessary to give a long abstract, but 16 lines in this paper are used to describe the comprehensive results of the propose method. Also, these results are described again in the Introduction. The authors should make the paper more succinct to fit the latex template.

2. For eq(3), the authors use LogDet matrix divergence to force the $X_{t}^{-1}$ and $X^{-1}_{t-1}$ to be "close". However, how the such closeness in LogDet matrix divergence could ensure the final term which is a complicated matrix multiplication in Eq3 to be safely omitted? Could the authors provide more details?

3. The authors could compare with more second-order approaches, such as K-FAC [36], K-BFGS [21], SON [33].

4. It would be highly recommended that the authors would release the code. Due to lack of details in the experiment, it may be hard to reproduce the reported results. For example, what is this large language model with 1B parameters used in the experiment?

5. Line 135. The authors seem to omit the item $c_t$ in the presented formula, despite that this item will not make influence regarding this optimization.

**Questions:**

See weakness.

**Limitations:**

I have not found any discussions about the limitations and potential negative societal impact. But in my opinion, this may not be a problem, since the work only focuses on the learning algorithm. Still, it is highly encouraged to add corresponding discussions.

---

> ### Author Rebuttal · Authors · 2023-08-10
>
> > The authors seem to have largely changed the spacing between sections. The current spacing presented in the paper would be much more narrow than that in the latex template. Slight changes in the latex template could be accecptable, but such large changes could not be available. I could understand that the authors try to present as much content as possible. But the writing seems to be quite wordy. For example Abstract contains 24 lines, which btw is near 15 lines for other papers in common. I understand sometimes it is necessary to give a long abstract, but 16 lines in this paper are used to describe the comprehensive results of the propose method. Also, these results are described again in the Introduction. The authors should make the paper more succinct to fit the latex template.”
>
> Thanks for bringing to our notice. We promise to fix the formatting in the final draft.
>
> > For eq(3), the authors use LogDet matrix divergence to force the  and to be "close". However, how the such closeness in LogDet matrix divergence could ensure the final term which is a complicated matrix multiplication in Eq3 to be safely omitted? Could the authors provide more details?
>
> Logdet Divergence between $X_t^{-1}$ and $X_{t-1}^{-1}$ is used as a regularizer, since the lower the logdet divergence leads to low value of the final term in eq (3). Since $w^*$ is unknown LogDet divergence serves as a proxy for this term.
>
> > The authors could compare with more second-order approaches, such as K-FAC [36], K-BFGS [21], SON [33].”.
>
> Thanks for the comment. We compared with rfdSON in the paper. And in addition, we also compare with KFac, FishLeg, and Eva on the autoencoder benchmark. Please refer to the rebuttal pdf attached.
>
> > It would be highly recommended that the authors would release the code. Due to lack of details in the experiment, it may be hard to reproduce the reported results. For example, what is this large language model with 1B parameters used in the experiment?
>
> We are sharing our code though an anonymized link to the ACs, since sharing with reviewers isn't allowed now. The 1B model used is of Primer architecture [43], as also mentioned in the paper.
>
> >“Line 135. The authors seem to omit the item  in the presented formula, despite that this item will not make influence regarding this optimization
>
> Thanks for pointing this out, we’ll correct this typo in the revision.
>
> > I have not found any discussions about the limitations and potential negative societal impact. But in my opinion, this may not be a problem, since the work only focuses on the learning algorithm. Still, it is highly encouraged to add corresponding discussions.”
>
> We promise to add a discussion, further talking about the limitation of our work.
>
> -----
> We hope that the rebuttal clarifies questions raised by the reviewer. We would be very happy to discuss any further questions about the work, and would really appreciate an appropriate increase in score if reviewers’ concerns are adequately addressed to facilitate acceptance of the paper.

---

> > ### Comment · Reviewer_rRBk · 2023-08-12
> > **Thanks for the authors' kindly response.**
> >
> > Thanks for the authors' kindly response. Overall, I think the proposed method is interesting and promising, and may have further impact in regards to the area of learning methods. The authors promise to make revisions in their final version to address the concerns. I recommend for acceptance.

---

### Official Review · Reviewer_jEyd · 2023-07-11

**Soundness:** 2 fair
**Presentation:** 3 good
**Contribution:** 2 fair
**Rating:** 5
**Confidence:** 3

**Summary:**

The paper proposes a new second-order method, named SONew, to optimize convex problems theoretically and conducts experiments on non-convex deep learning training tasks, which show SONew trains faster than first-order Adam and second-order fsdSON and Shampoo in some scenarios like the ViT benchmarks. The key idea of SONew is an online Newton method that uses an approximate LogDet divergence with a sparisified preconditioner. The sparse approximation of the LogDet divergence allows the efficient computation of second-order information so that SONew is compute-efficient enjoying exploiting some second-order information to accelerate the training.  Experiments were conducted on various DNNs like Autoencoders, ViT, GNNs, and LLMs, showing SONew’s possible acceleration over some existing first-order methods and second-order methods.

**Strengths:**

1. The proposed sparsified approximation of LogDet divergence is a new direction to reduce the computing cost using second-order information in optimizing DNNs.
2. A theoretical analysis is provided to give an insight for the sparsified preconditioner.

**Weaknesses:**

1. The theoretical analysis is particularly for convex problems, so it may not hold as a DNN optimizer under non-convex optimization.
2. How to choose the sparse graph is unknown. The paper doesn’t conclude what sparse graphs should be used for different DNNs.
3. Experimental results show that SONew is worse than existing second-order methods like Shampoo (e.g., Table 1). In addition, some recent compute- and memory-efficient second-order methods (e.g., M-FAC [A] and Eva [B]) were not included in the comparison.

[A] M-FAC: Efficient Matrix-Free Approximations of Second-Order Information, NeurIPS 2021.
[B] Eva: Practical Second-order Optimization with Kronecker-vectorized Approximation, ICLR 2023.

**Questions:**

1. In the experimental results in Section 5, some results show that SONew is obviously worse than Shampoo, e.g., Fig. 1(b), Fig. 2(b) in terms of iterations to loss (or accuracy). How to conclude that SONew achieves “up to 30% faster convergence, 3.4% relative improvement in validation performance, and 80% relative improvement in training loss” in the abstract? How about comparing to existing compute-efficient and/or memory-efficient second-order methods like M-KFAC and Eva?

2. Given a deep neural network for training, how we can choose tridiag-SONew, band-4-SONew, band-8-SONew, or band-$n$-SONew to achieve good training performance?

3. The paper provides a regret bound for SONew in the online convex optimization framework, but the optimizer is particularly used in DNNs which are known non-convex problems. How is the established bound used in DNN optimization?

[A] M-FAC: Efficient Matrix-Free Approximations of Second-Order Information, NeurIPS 2021.
[B] Eva: Practical Second-order Optimization with Kronecker-vectorized Approximation, ICLR 2023.

**Limitations:**

See above.

---

> ### Author Rebuttal · Authors · 2023-08-10
>
> > How to choose the sparse graph is unknown. The paper doesn’t conclude what sparse graphs should be used for different DNNs.
>
>  Thanks for the question. Currently we developed SONew primarily for banded diagonal sparsity. That said, exploring other sparse graphs and choosing among them, is a future work.
> > Experimental results show that SONew is worse than existing second-order methods like Shampoo (e.g., Table 1). In addition, some recent compute- and memory-efficient second-order methods (e.g., M-FAC [A] and Eva [B]) were not included in the comparison.
>
>  We present a memory efficient optimizer. Hence, a fair comparison would be with other methods using memory linear in the number of parameters. Among all such methods, SONew performs significantly better. Moreover, taking your comments into consideration, we have included comparison with Eva on the Autoencoder benchmark in the additional rebuttal pdf. Please note due to computational and time budget constraints, we preferred comparing to Eva over MFac as authors show in the Eva paper that it performs better than KFac.
>
> > In the experimental results in Section 5, some results show that SONew is obviously worse than Shampoo, e.g., Fig. 1(b), Fig. 2(b) in terms of iterations to loss (or accuracy). How to conclude that SONew achieves “up to 30% faster convergence, 3.4% relative improvement in validation performance, and 80% relative improvement in training loss” in the abstract? How about comparing to existing compute-efficient and/or memory-efficient second-order methods like M-KFAC and Eva?”
>
>  By that statement, we meant in comparison to other optimizers which are memory efficient. We’ll clarify this in the final draft Abstract. We also compare with Eva, as mentioned previously.
>
> > The paper provides a regret bound for SONew in the online convex optimization framework, but the optimizer is particularly used in DNNs which are known non-convex problems. How is the established bound used in DNN optimization?”
>
>
> Our analysis is on regret minimization in online convex optimization which has several connections to non-convex optimization a) [1] uses OCO learners for optimizing non-convex objectives b) similarly [2] shows transferability of regret bound analysis to non-convex non-smooth objectives.  We also briefly discuss this in A.2.5
>
> [1] N. Agarwal, R. Anil, E. Hazan, T. Koren, and C. Zhang. Learning rate grafting: Transferability of optimizer tuning, 2022
>
> [2] Ashok Cutkosky, Harsh Mehta, and Francesco Orabona. Optimal stochastic non smooth non-convex optimization through online-to-non-convex conversion. arXiv preprint arXiv:2302.03775, 2023.
>
> ------
> We hope that the rebuttal clarifies questions raised by the reviewer. We would be very happy to discuss any further questions about the work, and would really appreciate an appropriate increase in score if reviewers’ concerns are adequately addressed to facilitate acceptance of the paper.

---

> > ### Author Response · Authors · 2023-08-15
> > **Official Comment by Authors**
> >
> > We hope that the rebuttal clarifies questions raised by the reviewer. Please let us know if we can add more to justify the contributions and significance of our work. We kindly request you to reconsider your rating if our responses were sufficiently able to address your concerns.

---

> > > ### Comment · Reviewer_jEyd · 2023-08-18
> > >
> > > Thanks for the clear clarification and additional results. I raise my score accordingly.

---

### Official Review · Reviewer_3rTc · 2023-07-21

**Soundness:** 3 good
**Presentation:** 4 excellent
**Contribution:** 2 fair
**Rating:** 5
**Confidence:** 4

**Summary:**

This paper proposes a memory-efficient and scalable second-order optimizer derived from the LogDet matrix divergence measure combined with sparsity constraints. The resulting preconditioner admits a banded/tridiagonal structure found via an efficient algorithm.

**Strengths:**

- The paper is well-written and fairly easy to follow. The steps are clear and logically laid out.

- I find the derivation of the preconditioner from the LogDet divergence interesting, even though variations of the particular form has been explored before.

- The theoretical framework around the proposed method looks sound and the analysis of numerical stability is appreciated.

**Weaknesses:**

- The banded/tridiagonal structure for which an efficient solution exists might not be natural, depending on the model architecture. A block-diagonal pattern such as in KFAC that models inter-layer dependencies would potentially be more suitable.

- The paper is positioned as a practical, scalable optimizer for large-scale deep learning tasks. As such, I am not completely convinced by the experimental results. See breakdown below.

- First, comparison with KFAC in terms of generalization and wall-clock performance would be important, as it is one of the very few higher order optimizers in deep learning that has shown adoption in practice.

- Based on the presented experiments, it is hard to tell when one would pick the proposed optimizer over a simple Adam. Training loss value is not necessarily relevant, as second-order optimizers are known to often quickly converge to local optima with low loss but poor generalization. The plot showing validation error rate does not depict how this translates to classification accuracy. In case of minor/negligible differences one would pick Adam that offers less added complexity and faster wall-clock performance. Overall, there could be more focus on wall-clock performance, as performance with respect to number of steps is not that relevant in practice.

**Questions:**

- How does the method compare with KFAC focusing on generalization on standard tasks explored in the paper and with respect to wall-clock time?

- What is the practical advantage of the proposed method compared with well-established optimizers such as Adam? Can the proposed method accelerate training speed, in terms of time to achieve SOTA accuracies? If not, how much does generalization suffer compared with SOTA first-order optimizers?

- Minor comments:

  - The notation for indexing the matrix in Eq. 12 is confusing.

  - Personal preference: I wouldn't use subjective terms such as embarassingly parallelizable or beautiful paper.

**Limitations:**

Some limitations are discussed.

---

> ### Author Rebuttal · Authors · 2023-08-10
>
> We thank the reviewer for their comments.
>
> >The banded/tridiagonal structure for which an efficient solution exists might not be natural, depending on the model architecture. A block-diagonal pattern such as in KFAC that models inter-layer dependencies would potentially be more suitable.
>
> One can indeed capture inter-layer dependencies as well. There are many ways to do this. For example, one can think of all the parameters as one set rather than viewing them as a set of parameters from different layers. This way, there’ll be cross correlations across parameters of different layers which can be captured. We leave testing this as a future work.
>
> > comparison with KFAC in terms of generalization and wall-clock performance would be important, as it is one of the very few higher order optimizers in deep learning that has shown adoption in practice.
>
> - We compare optimizers via thorough hyperparameter search comprising several candidate hyperparameter configurations, hence given the limited resources we picked shampoo from the class of kronecker factored optimizers (KFAC, KBFGS), given that shampoo’s empirical performance is similar to that of KFAC and KBFGS as demonstrated in Shampoo paper [4].
> - Taking your comment into consideration, we also run additional Autoencoder benchmark experiment comparing with KFac, FishLeg, and Eva. The plot is provided in the additional rebuttal pdf document. We notice that SONew beats all other baselines by a significant margin.
>
> > “Based on the presented experiments, it is hard to tell when one would pick the proposed optimizer over a simple Adam. Training loss value is not necessarily relevant, as second-order optimizers are known to often quickly converge to local optima with low loss but poor generalization. The plot showing validation error rate does not depict how this translates to classification accuracy.”
>
> When one can afford “num of params” extra memory, SONew finds the best use as demonstrated in our experiemnts. If there’s a strict memory bottleneck, one can prefer Adam.
> Just to clarify validation error rate is = 1 - validation accuracy, and since it is measured in a held out dataset which is not used in training and hence the validation error rate measures the generalization performance of the model and in practice since we don’t have test data, one always uses validation/development/held-out dataset to choose the best generalizing model.
>
> > there could be more focus on wall-clock performance, as performance with respect to number of steps is not that relevant in practice.
>
> Thanks for the comment. – (already written above)
>
> > "How does the method compare with KFAC focusing on generalization on standard tasks explored in the paper and with respect to wall-clock time?”
>
>  In the limited time of the rebuttal period, we could set up and run additional baselines including KFac, FishLeg, and Eva only on the Autoencoder benchmark. These comparisons are done wrt wall-clock time. We will further test these baselines on more benchmarks and include the results in our final draft.
>
> > What is the practical advantage of the proposed method compared with well-established optimizers such as Adam? Can the proposed method accelerate training speed, in terms of time to achieve SOTA accuracies? If not, how much does generalization suffer compared with SOTA first-order optimizers?
>
>  SONew significantly improves over the training time while taking extra memory only linear in the number of parameters. Specifically, we notice upto 30% faster convergence in OGBG, 10% in ViT, and upto 26% on LLMs.
>
> >notation for indexing the matrix in Eq. 12 is confusing.
>
>  Thanks We’ll fix the formatting.
>
> > Personal preference: I wouldn't use subjective terms such as embarassingly parallelizable or beautiful paper.”
>
>  Thanks. We’ll keep this in mind and edit such occurrences.
>
> -------
> We hope that the rebuttal clarifies questions raised by the reviewer. We would be very happy to discuss any further questions about the work, and would really appreciate an appropriate increase in score if reviewers’ concerns are adequately addressed to facilitate acceptance of the paper.

---

> > ### Author Response · Authors · 2023-08-15
> > **Official Comment by Authors**
> >
> > We hope that the rebuttal clarifies questions raised by the reviewer. Please let us know if we can add more to justify the contributions and significance of our work. We kindly request you to reconsider your rating if our responses were sufficiently able to address your concerns.

---

> > ### Comment · Reviewer_3rTc · 2023-08-16
> >
> > Thank you for addressing my questions and providing additional experiments. I believe the work is interesting and shows merit. On the other hand, the gains are very minor in most experiments, and the added complexity compared to Adam might not worth it in most practical scenarios. I am increasing my score to 5 to reflect that most of my issues have been addressed.

---

### Official Review · Reviewer_UQcK · 2023-07-22

**Soundness:** 2 fair
**Presentation:** 2 fair
**Contribution:** 2 fair
**Rating:** 5
**Confidence:** 4

**Summary:**

This paper introduces an online Newton method for optimization of convex losses, with the goal of applying it to large neural networks.
In particular, the algorithm is derived by optimizing a preconditioning matrix on a regret bound and using the logdet divergence as a regularizer.
It introduces a sparsification procedure to alleviate the computational burden of computing and inverting the full pre-conditioning matrix.
With a fixed spasification structure (banded) of the preconditioning matrix, the problem regularized by the logdet divergence admits solutions that can be computed efficiently.
The method is applied to a variety of (non-convex) benchmarks and compared with other optimizers.

**Strengths:**

1) Fast optimization of neural networks is very important, to possibly reduce the amount of resources necessary for training neural networks.

2) Theoretical derivation of the algorithm is clear and sound.


**Weaknesses:**

1) The derivation of the algorithm is based on the assumption of a convex loss function.
Most of previous work discussed in the "Related work" section is about other online Newton methods, also designed for convex problems.
However, all numerical experiments are shown on non-convex problems.
This is problematic, because if the claim is that the method should be used for second-order optimization of non-convex losses, then it should be compared with the corresponding benchmarks: KFAC, K-BFGS, FishLeg.
Indeed, the paper acknowledges that KFAC uses a similar sparsification structure (tridiagonal), but no empirical comparison is provided.
On the other hand, if the claim is that the method should be used for convex optimization, then it should be clarified what are the (theoretical and empirical) advantages of the algorithm with respect to other online Newton methods.

2) Some of the statements made in the paper are misleading. For example:
i) The algorithms provided on page 5 are not actually the algorithms used for the numerical experiments.
"Adam grafting" is used on top of those algorithm, but no justification is given, also in the light of the proposed theory and derivations.
My assumption is that none of the theory developed holds after including "Adam grafting" in the algorithm, and a description of the full algorithm should be given instead of just giving a reference to another paper.
ii) The following statement in section 5 is not true: "We compare SONew against widely used first order methods including SGD, SGD with Momentum, Nesterov, Adagrad, Adam, and Rmsprop."
The first four of the six optimizers mentioned in the list are actually not compared with.
iii) The abstract says that the proposed method is slower than first-order methods, but no results are shown in the results about this.
In fact, figures show that the proposed method is faster in terms of number of steps, but what is the significance of this results if the method is overall slower?

3) The results of numerical experiments are underwhelming.
As I already mentioned, no comparison with other second-order methods is given, besides Shampoo, and comparison with simple and strong benchmarks is also not provided.
For example, SGD with momentum is never compared with, and neither Adam nor SGD are compared with in the case of LLM in figure 3.
Furthermore, it is not clear if the method provides a reliable advantage.
Only single training runs are shown, and fluctuations look pretty large in all figures.
Averaging over multiple training runs would help understanding how reliable is the method.
Finally, training curves are shown only vs epochs or number of steps, no results in terms of wall clock time are provided, therefore it is hard to tell whether the method provides any real advantage.


**Questions:**

See above

**Limitations:**

See above

---

> ### Author Rebuttal · Authors · 2023-08-10
>
> We thank the reviewer for their comments.
>
> > The derivation of the algorithm is based on the … corresponding benchmarks: KFAC, K-BFGS, FishLeg
>
> There are multiple reasons behind our choice of the baselines for each benchmark. We explain them here:
>
> - Our aim is to present a memory efficient second order optimizer, hence targeted for large models as commonly used in deep learning. While KFAC, FishLeg are popular second order methods, they require $O(d_1^2 + d_2^2)$ memory, which can be significantly larger than $O(d1*d2)$ in the context of deep learning models. Hence, we rather compare with first order methods and a memory efficient second order optimizer rfdSON, as they take similar memory
>
> - We compare optimizers via thorough hyperparameter search comprising several candidate hyperparameter configurations, hence given the limited resources we picked shampoo from the class of kronecker factored optimizers (KFAC, KBFGS), given that shampoo’s empirical performance is similar to that of KFAC and KBFGS as demonstrated in Shampoo paper [4].
>
> - Furthermore as per your suggestion, we compared against KFAC, FishLeg and Eva (a recently proposed memory efficient second order method) on Autoencoder benchmark. We make the comparison with respect to wall clock time and notice that SONew beats all the baselines by a large margin. The plot is provided in the pdf attached to rebuttal.
>
> > The algorithms ... to another paper
>
> It is a common practice in second order algorithms to ensure that the preconditioned gradient/update norm is in a trust region [EVA]. For example, i) update clipping in EVA, KFAC ii) grafting in shampoo [4], which ensures that the final update norm is < d, where d is the dimension of the update. In this work we chose grafting and leave exploring other ways to control the update norms as future work.
>
> [4] Anil et al. 2021 Scalable second order optimization for deep learning
>
> [EVA] Eva: Practical Second-order Optimization with Kronecker-vectorized Approximation, ICLR 2023
>
> > The following statement in section 5 is not true: "We compare SONew against widely used first order methods including SGD, SGD with Momentum, Nesterov, Adagrad, Adam, and Rmsprop." The first four of the six optimizers mentioned in the list are actually not compared with.
>
> Due to space constraints we deferred the comparison with the first four to Section A.4.3 as they were least performing among the six (as mentioned in Line 285-286)
>
> > The abstract says that the proposed method is slower ... overall slower?
>
> In all the experiments, we notice that SONew is indeed slower than first order methods, though by a very small margin (<=3% relative overhead). Though, as you rightly pointed out - it gives significant gains overall (and hence a useful method) as it converges faster. We’ll clarify this in the abstract.
>
> > The results of numerical experiments are underwhelming. As I already mentioned, no comparison with other second-order methods is given, besides Shampoo
>
> As previously mentioned, given the computation budget, we compared with only the strongest baselines like Adam and Shampoo for large scale benchmarks. Though, as per your request, we have added comparison with FishLeg, KFac, and Eva  on Autoencoder benchmark in the additional rebuttal pdf.
>
> > For example, SGD with momentum is never compared with, “
>
> We compare Momentum and several other first order methods on Autoencoder benchmark where we noticed they perform significantly suboptimal compared to baseline like Adam. Hence, for larger benchmarks like ViT and OGBG, we fixed Adam as the first order baseline to compare with. Though, as per your request, we’ve now added SGD+Momentum and RmsProp as well to ViT and OGBG benchmark (provided in the additional pdf).
>
> > Adam nor SGD are compared with in the case of LLM in figure 3”
>
> LLM requires a very high computational budget, hence it was not possible to add simple baseline like SGD which will surely perform poorer compared to Adam. We indeed do compare with Adam. The AdaFactor optimizer we use in LLM experiments is without factoring, which resorts to Adam like update with some additional steps like update clippings. The reason to choose AdaFactor (with no factoring) rather than Adam is it has been seen in past that to train LLMs, AdaFactor is an optimal choice (as mentioned in the paper) [10, 46]
>
> [10]Palm: Scaling language modeling with pathways, 2022
>
> [46] T. Wang, A. Roberts, D. Hesslow, T. L. Scao, H. W. Chung, I. Beltagy, J. Launay, and C. Raffel. What language model architecture and pretraining objective work best for zero-shot generalization, 2022.
>
> > Finally, training curves are shown only vs epochs or number of steps, no results in terms of wall clock time are provided, therefore it is hard to tell whether the method provides any real advantage
>
> - Our work presents a memory efficient second optimizer. Hence, a fair comparison would involve optimizers using similar memory, which happens to be first order methods or rfdSON. But for all these methods, the running time are within 5% relative overhead of each other. Hence, comparing against time is as good as comparing against steps. That said, we take your comments and also run Shampoo on OGBG (It couldn’t scale to ViT and LLM) for similar amount of time. We find that SONew performs as good as Shampoo though using significantly less memory! Thanks for your suggestion.
>
> - Additional experiment on Autoencoder benchmark with KFac, FishLeg, and Eva are performed optimizing with respect to a common wall-clock time. The plots can be foud in additional pdf attached to the rebuttal. We find that SONew beats other baselines by a great margin.
>
>
> --------
> We hope that the rebuttal clarifies questions raised by the reviewer. We would be very happy to discuss any further questions about the work, and would really appreciate an appropriate increase in score if reviewers’ concerns are adequately addressed to facilitate acceptance of the paper.

---

> > ### Author Response · Authors · 2023-08-15
> > **Official Comment by Authors**
> >
> > We hope that the rebuttal clarifies questions raised by the reviewer. Please let us know if we can add more to justify the contributions and significance of our work. We kindly request you to reconsider your rating if our responses were sufficiently able to address your concerns.

---

> > ### Comment · Reviewer_UQcK · 2023-08-16
> >
> > Thank you for providing a more detailed comparison. I will increase the score. However I still think that, to avoid confusion, the paper should state very clearly that, while the theory is derived for convex loss functions, it is tested on non-convex losses, and provide clear justifications for this choice (which still remains unclear to me even after the authors' reply)

---

### Official Review · Reviewer_KiQx · 2023-07-25

**Soundness:** 3 good
**Presentation:** 4 excellent
**Contribution:** 4 excellent
**Rating:** 7
**Confidence:** 3

**Summary:**

This manuscript proposes SONew, a second-order algorithm that uses sparsity in the preconditioner to distil to linear time and space complexity. Different sparsity patterns (tridiagonal, banded, etc.) allow generalization of the algorithm for training deep neural networks almost as efficiently as first-order methods. SONew achieves up to 30% faster convergence compared to Adam and improves validation performance when trained for the same number of iterations when training with the Adam optimizer.

**Strengths:**

•	The paper provides a strong theoretical foundation for sparsifying the preconditioning matrix, making a clear connection between regret minimization and  LogDet divergence of the preconditioning matrix
•	The paper provides an extensive suite of experiments indicating the computational efficiency and performance gains of their method on large scale model training
•	The authors provide theoretical guarantees of the numerical stability of their method, which is a serious concern for existing second-order approximation OCO algorithms
•	The paper is clearly written


**Weaknesses:**

The paper does a good job providing both theoretical and empirical justification for their method. However, there are a few details about the experimental setup and bottlenecks for baselines that would improve the manuscript. The limitations section should also be expanded to explain under what conditions SONew would not be optimal or theoretical guarantees would fail. Details provided below in the questions section:

**Questions:**

•	The authors compare Shampoo with RMSProp grafting but use Adam grafting for their method. Adam grafting would yield a more accurate step size compared to RMSProp. It would be beneficial to add a baseline of Shampoo with Adam grafting
•	Fig. 1a mentinos that hyperparameter search could not be performed on Shampoo on 16GB TPU. Could the precision be reduced (bfloat16) to run shampoo?


**Limitations:**

The authors need to expand on the limitation of their method and point out specific cases where their method may be disadvantaged compared to existing methods.

---

> ### Author Rebuttal · Authors · 2023-08-10
>
> We thank the reviewer for the comments
>
> > “The authors compare Shampoo with RMSProp…”
>
> We compare Shampoo using RmsProp grafting since that is the default value of grafting hyperparameter set. Moreover, Adam grafting would require an additional $O(NumParams)$ values to be stored to maintain the exp moving average of the diagonals, leading to extra overhead over already large memory overhead by Shampoo.
>
> > “Fig. 1a mentions that hyperparameter search..”
>
> Thanks for the suggestion. Indeed one can use bfloat16 and test Shampoo using that. Though as we move to even larger benchmarks, converting to bfloat16 precision still won’t scale. So the memory bottleneck issue of Shampoo still exists. We leave bfloat16 experiments of Shampoo, as well as SONew as a future work.
>
> > “The authors need to expand on the limitations of their method and ..”
>
> We’ll surely expand upon the limitations and include specific cases where SONew might not work. For example, when the activations are sparse, then SONew might not be capturing much useful information as two consecutive neurons might not be firing up simultaneously a lot.

---

> > ### Comment · Reviewer_KiQx · 2023-08-12
> >
> > Thanks for the authors' kindly response. Overall, I think the proposed method is interesting and promising, and may have further impact in regards to the area of learning methods. The authors promise to make revisions in their final version to address the concerns. I recommend for acceptance.
> >
> > I appreciate the authors' response. Regarding the points above
> > 1. Using Adam grafting with shampoo would help ablate the importance of grafting type to different optimizer settings
> > 2. mixed precision training (including w/ bfloat16) is the norm in LLM training. this would substantially improve comparisons regarding memory
> > 3. it's helpful that the authors listed an additional limitation, but it is important that the final version has a deeper dive into the limitations and potential impact of these limitations.
> >
> > b/c of these points i am maintaining my score

---

### Official Review · Reviewer_XomD · 2023-07-29

**Soundness:** 3 good
**Presentation:** 3 good
**Contribution:** 3 good
**Rating:** 6
**Confidence:** 1

**Summary:**

This paper propose a memory-efficient second-order algorithm named Sparsified Online Newton (SONew) method. By introducing the LogDet matrix divergence for regret analysis, the authors derives an online Newton update rule. Then the authors introduces structured sparsity patterns on the preconditioner matrix on the preconditioner matrix to reduces storage and computational complexity. They adopt the proposed methods on several benchmarks and show SOTA performances comparing to other optimization methods.

**Strengths:**

1. The writing and organization of the paper are very good, the notation representation is accurate.
2. The method proposed in the paper is reasonable, and the paper applies the proposed method on some benchmarks, achieving better results compared to other optimizers.

**Weaknesses:**

1. As an optimization method with lower resource usage, the authors only mention the theoretical reduction in space complexity, and the actual decrease in training time compared to other optimizers. However, they do not provide actual comparisons of GPU memory usage.
2. The authors evaluate the proposed optimization method on deep autoencoders, Vision Transformers, and Graph Neural Networks. However, I think that as a newly proposed optimization method, the authors need to conduct experiments on more benchmarks and more network architectures to verify its general applicability, such as results of more CNN methods on ImageNet, and generative models in computer vision like Diffusion Models.

**Questions:**

1. Could the author provide the actual GPU memory cost and comparing the proposed method with other methods?
2. Could the author provide more experiments to show the generalization of the proposed methods?

**Limitations:**

The author does not discuss the limitations and potential negative societal impact of their work. I consider one of the limitations of the proposed method is that it can not be guaranteed that this method can achieve better performance compared to other optimizers on more datasets and network architectures.

---

> ### Author Rebuttal · Authors · 2023-08-10
>
> We thank the reviewer for their comments.
>
> > Could the author provide the actual GPU memory cost and comparing the proposed method with other methods?
>
> We mention these numbers for ViT on Shampoo (155M optimizer params), SONew (44M optimizer params), and Adam (30M optimizer params). We promise to report similar comparison for all other baselines and benchmark in the final draft
>
> >Could the author provide more experiments to show the generalization of the proposed methods?
>
> We included additional baseline in the rebuttal pdf to show SONew's performance with respect to other second order methods. We promise to include more large scale experiments in addition to the existing ones in the final draft.

---

> > ### Comment · Reviewer_XomD · 2023-08-16
> >
> > Thanks for your response.  I suggest the author provide a table to illustrate the memory cost of different optimizers in the final draft.
> > I vote for accept and I am maintaining my score.

---

> > > ### Author Response · Authors · 2023-08-21
> > > **Official Comment by Authors**
> > >
> > > We thank the reviewer for the follow up. We provide here an estimate of the optimizer parameters count in terms of the number of parameters of the network, for different benchmarks and different baselines used in our experiments:
> > >
> > > Let the number of parameters of the network be $n$.
> > >
> > > Benchmark: Autoencoder ($n \sim 1.4M$)
> > >
> > > | Optimizer  | K-FAC | Shampoo       | Fishleg    | Eva   | Adam | SGD+Momentum | RmsProp | tds-SONew |
> > > | --- | --- | --- | --- |--- | --- |--- | --- |  --- |
> > > |number of opt params | $5.56n$   | $6.56 n$        | $4.28 n$ | $n$ 	| $2n$    | $n$ 	     | $n$ 	  | $3n$ 	|
> > >
> > >
> > > Benchmark: GraphNetwork ($n \sim 3.5M$)
> > >
> > > | Optimizer      | Shampoo | Adam | SGD+Momentum | RmsProp | tds-SONew |
> > > | --- | --- | --- | --- |--- | --- |
> > > |number of opt params | $11n$          | $2n$       | $n$    | $n$		|	 $3n$      |
> > >
> > > Benchmark: ViT ($n \sim 22M$)
> > >
> > > |Optimizer      | Shampoo | Adam | SGD+Momentum | RmsProp | tds-SONew |
> > > | --- | --- | --- | --- |--- | --- |
> > > |number of opt params | $7n$            | $2n$       | $n$                           | $n$		|	 $3n$      |
> > >
> > > Benchmark: Language Model ($n \sim 1B$)
> > >
> > > |Optimizer       |Adam |tds-SONew |
> > > | --- | --- | --- |
> > > | number of opt params | $2n$       |	 $3n$      |
> > >
> > > We’ll also provide exact numbers in the final draft in addition to the above.
> > >
> > > -----
> > >
> > > We hope that the rebuttal clarifies questions raised by the reviewer. We would be very happy to discuss any further questions about the work, and would really appreciate an appropriate increase in score if reviewers’ concerns are adequately addressed to facilitate acceptance of the paper.

---

### Author Rebuttal · Authors · 2023-08-10

Common Rebuttal:
- As Reviewers suggested, we include additional experiment on the Autoencoder benchmark in the pdf attached. We compare SONew with KFac [1], Eva [2], and FishLeg [3]. As these baselines were not available in JAX, we used a PyTorch version as available in their official github repositories. Furthermore, as reviewers mentioned we make wall-clock comparison with these new baselines. Note that since the experiment setup is a bit different, the results of SONew don't match the ones presented in the main paper conducted using JAX. One main reason being we run 600 hyperparam configs for these experiments, and for the Autoencoder experiments presented in the main paper we do 2000 hyperparam configs.

-  Comparison in Wall-clock time: Reviewers mentioned the need to have wall clock time comparison. We want to highlight that SONew performs similar in per step time (<=2% relative overhead) compared to other memory efficient optimizers, which mostly include first order methods or rfdSON. Therefore, in a comparison with other methods using memory only linear in the number of parameters, comparison in wall-clock time will be similar to comparison in steps. Furthermore, we make a wall-clock time comparison with Shampoo in the additional pdf and find SONew performs better than Shampoo while using significantly less memory.

[1] Martens et. al, 2015 Optimizing Neural Networks with Kronecker-factored Approximate Curvature

[2] Eva: Practical Second-order Optimization with Kronecker-vectorized Approximation, ICLR 2023

[3] Gracia et al., 2023 Fisher-Legendre (FishLeg) optimization of deep neural networks.

---

### Decision · Program_Chairs · 2023-09-21

**Decision:**

Accept (poster)

**Comment:**

Motivated by finding scaling matrices that minimize the regret bound in online convex setting, the paper introduces a scaled gradient algorithm that by leveraging the properties of LogDet matrix divergence coupled with sparsity constraints offers a great deal of memory-efficiency. The paper is well written and the theoretical derivations are solid. The numerical experiments can be improved, however, to include more models and alternative methods.